# Unmasking the Lottery Ticket Hypothesis: What's Encoded in a Winning Ticket's Mask?

**Mansheej Paul**[1,2*] **Feng Chen**[1*] **Brett W. Larsen**[1,3*] **Jonathan Frankle**[4,5] **Surya Ganguli**[1,2] **Gintare Karolina Dziugaite**[6,7]
[1]Stanford, [2]Meta AI, [3]Flatiron Institute, [4]MosaicML, [5]Harvard, [6]Google Research, Brain Team, [7]Mila; McGill

## Abstract

As neural networks get larger and costlier, it is important to find sparse networks that require less compute and memory but can be trained to the same accuracy as the full network (*i.e. matching*). Iterative magnitude pruning (IMP) is a state of the art algorithm that can find such highly sparse *matching subnetworks*, known as *winning tickets*. IMP iterates through cycles of training, pruning a fraction of smallest magnitude weights, rewinding unpruned weights back to an early training point, and repeating. Despite its simplicity, the principles underlying when and how IMP finds winning tickets remain elusive. In particular, what useful information does an IMP mask found at the *end* of training convey to a rewound network near the *beginning* of training? How does SGD allow the network to extract this information? And why is *iterative* pruning needed, i.e. why can't we prune to very high sparsities in one shot? We investigate these questions through the lens of the geometry of the error landscape. First, we find that—at higher sparsities—pairs of pruned networks at successive pruning iterations are connected by a linear path with zero error barrier if and only if they are matching. This indicates that masks found at the end of training convey to the rewind point the identity of an axial subspace that intersects a desired linearly connected mode of a matching sublevel set. Second, we show SGD can exploit this information due to a strong form of robustness: it can return to this mode despite strong perturbations early in training. Third, we show how the flatness of the error landscape at the end of training limits the fraction of weights that can be pruned at each iteration of IMP. This analysis yields a new quantitative link between IMP performance and the Hessian eigenspectrum. Finally, we show that the role of retraining in IMP is to find a network with new small weights to prune. Overall, these results make progress toward demystifying the existence of winning tickets by revealing the fundamental role of error landscape geometry in the algorithms used to find them.

## 1 Introduction

Recent advances in deep learning have been driven by massively scaling both the size of networks and datasets (Kaplan et al., 2020; Hoffmann et al., 2022). But this scale comes at considerable resource costs. For example, when training or deploying networks on edge devices, memory and computational demands must remain small. This motivates the search for sparse trainable networks (Blalock et al., 2020), pruned datasets (Paul et al., 2021; Sorscher et al., 2022), or both (Paul et al., 2022) that can be used within these resource constraints.

However, finding highly sparse, *trainable* networks is challenging. A state of the art algorithm for doing so is iterative magnitude pruning (IMP) (Frankle et al., 2020a). IMP starts with a dense network usually pretrained for a very short amount of time. The weights of this network are called the *rewind point*. IMP then repeatedly (1) trains this network to convergence; (2) prunes the trained network by computing a mask that zeros out a fraction (typically 20%) of the smallest magnitude weights; (3) rewinds the nonzero weights back to their values at the rewind point, and then starts the next iteration by training the masked network to convergence. Each successive iteration yields a mask with higher sparsity. The final mask applied to the rewind point constitutes a highly sparse trainable subnetwork called a *winning ticket* if it trains to the same accuracy as the full network, i.e. is *matching*.

---

*Equal contribution. Correspondence to: `mansheej@stanford.edu`; `gkdz@google.com`.

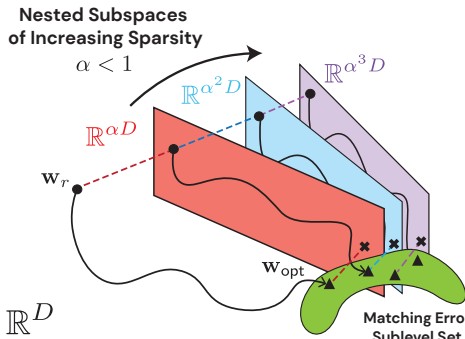

**Nested Subspaces of Increasing Sparsity**

$\alpha < 1$

$\mathbb{R}^{\alpha^3 D}$

$\mathbb{R}^{\alpha^2 D}$

$\mathbb{R}^{\alpha D}$

$\mathbf{w}_r$

$\mathbf{w}_{opt}$

$\mathbb{R}^D$

Matching Error Sublevel Set

Figure 1: Error landscape of IMP. At iteration $L$, IMP trains the network from a pruned rewind point (circles), on an $\alpha^L D$ dimensional axial subspace (colored planes), to a level $L$ pruned solution (triangles). The smallest $(1-\alpha)$ fraction weights are then pruned, yielding the level $L$ projection ($\times$'s) whose $0$ weights form a sparsity mask corresponding to a $\alpha^{L+1} D$ dimensional axial subspace. This mask, when applied to the rewind point, defines the level $L + 1$ initialization. Thus IMP moves through a sequence of nested axial subspaces of increasing sparsity. We find that when IMP finds a sequence of matching pruned solutions (triangles), there is no error barrier on the piecewise linear path between them. Thus the key information contained in an IMP mask is the identity of an axial subspace that intersects the connected matching sublevel set containing a well-performing overparameterized network.

Although IMP produces highly sparse matching networks, it is extremely resource intensive. Moreover, the principles underlying when and how IMP finds winning tickets remain quite mysterious. Thus, the goal of this work is to develop a scientific understanding of the mechanisms governing the success or failure of IMP. By doing so, we hope to enable the design of improved pruning algorithms.

The operation of IMP raises four foundational puzzles. **First**, the mask that determines which weights to prune is identified based on small magnitude weights at the *end* of training. However, at the next iteration, this mask is applied to the rewind point found *early* in training. Precisely what information from the end of training does the mask convey to the rewind point? **Second**, how does SGD starting from the masked rewind point extract and use this information? The mask indeed provides actionable information beyond that stored in the network weights at the rewind point—using a random mask or even pruning the smallest magnitude weights at this point leads to higher error after training (Frankle et al., 2021). **Third**, why are we forced to prune only a small fraction of weights at each iteration? Training and then pruning a large fraction of weights in one shot does not perform as well as iteratively pruning a small fraction and then retraining (Frankle & Carbin, 2018). Why does pruning a larger fraction in one iteration destroy the actionable information in the mask? **Fourth**, why does retraining allow us to prune more weights? A variant of IMP that uses a different retraining strategy (learning rate rewinding) also successfully identifies matching subnetworks while another variant (finetuning) fails (Renda et al., 2020). What differentiates a successful retraining strategy from an unsuccessful one?

**Understanding IMP through error landscape geometry.** In this work, we provide insights into these questions by isolating important aspects of the error landscape geometry that govern IMP performance (Fig. 1). We do so through extensive empirical investigations on a range of benchmark datasets (CIFAR-10, CIFAR-100, and ImageNet) and modern network architectures (ResNet-20, ResNet-18, and ResNet-50). Our contributions are as follows:

- We find that, at higher sparsities, *successive* matching sparse solutions at level $L$ and $L + 1$ (adjacent pairs of triangles in Fig. 1) *are* linearly mode connected (there is no error barrier on the line connecting them in weight space). However, the dense solution and the sparsest matching solution may not be linearly mode connected. The transition from matching to nonmatching sparsity coincides with the failure of this successive level linear mode connectivity. This answers our **first** question of what information from an IMP solution at the end of training is conveyed by the mask to the rewind point: it is the identity of a sparser axial subspace that intersects the linearly connected sublevel set containing the current matching IMP solution. IMP fails to find matching solutions when this information is *not* present. (Section 3.1)

- We show that, at higher sparsities, networks trained from rewind points that yield matching solutions exhibit a strong form of robustness. The linearly connected modes these networks train into are not only robust to SGD noise (Frankle et al., 2020a), but also to random perturbations of length equal to the distance between rewind points at successive levels (adjacent circles in Fig. 1). This answers our **second** question of how SGD extracts information from the mask: two pruned rewind points at successive sparsity levels are likely to navigate back to the same linearly connected mode yielding matching solutions (adjacent triangles Fig. 1). (Section 3.2)

- We develop an approximate theory, based on Larsen et al. (2022), that determines how aggressively one can prune a trained solution (triangle in Fig. 1) in any level in terms of the length of the pruning projection (distance to associated $\times$ in Fig. 1) and the Hessian eigenspectrum at the solution. This theory sheds light on why flatter error landscapes allows more aggressive pruning. Our theory quantitatively matches random pruning while we show that magnitude pruning can slightly outperform it. (Section 3.3)

- We explain the superior performance of magnitude pruning by showing that this method identifies flatter directions in the error landscape compared to random pruning, thereby allowing more aggressive pruning per iteration. Together, these results provide new quantitative connections between error landscape geometry and feasible IMP hyperparameters. They also answer our **third** question why we need iterative pruning: one-shot pruning to high sparsities is prohibited by the sharpness of the error landscape. (Section 3.3)

- We show that a fundamental role of retraining is to reequilibriate the weights of the network, i.e. find networks with new small weights amenable to further pruning. Successful retraining strategies such as weight and learning rate (LR) rewinding (Renda et al., 2020) both do this while finetuning (FT), an unsuccessful retraining strategy, does not. This answers our **fourth** question regarding why retraining allows further magnitude pruning. (Section 3.4)

Overall, our results provide new geometric insights into IMP, including when this method breaks down and why. We hope this understanding will lead to new pruning strategies that match the performance of IMP but with fewer resources. Indeed, a simple adaptive rule for choosing the pruning ratio at each level leads to nontrivial improvements (Appendix F). Furthermore, understanding the role of weight reequilibriation in retraining may inspire methods to achieve this property faster.

**Related work.** A number of studies have focused on finding matching subnetworks at initialization (e.g., Lee et al., 2019; Wang et al., 2020; Tanaka et al., 2020). Despite substantial progress, these methods do not find matching solutions at scale (Su et al., 2020; Frankle et al., 2021); the trained dense network seems to be an important ingredient for IMP (e.g., Zhou et al., 2019; Frankle et al., 2020a; Ramanujan et al., 2020; Sreenivasan et al., 2022). Another essential part of IMP is to iteratively prune a small fraction of the weights with periods of retraining between them. For both training without rewinding (Han et al., 2015), and lottery tickets with rewinding (Frankle et al., 2020a), one-shot magnitude pruning cannot find matching subnetworks of the same sparsity as iterative pruning. Alternative methods that do so also feature iterations of pruning and retraining (Renda et al., 2020; Savarese et al., 2020). Additionally Zhou et al. (2019) and Frankle et al. (2020b) investigate what information is necessary for finding winning tickets with many ablations of which weights to prune and how to reinitialize weights for retraining. We explore the importance of these ingredients in IMP through the lens of the error landscape geometry.

The closest related works to our results are Frankle et al. (2020a) and Evci et al. (2022). Both works consider linear mode connectivity between two networks of the same sparsity. Frankle et al. (2020a) compare two networks trained from the same rewind point at the same pruning level but with different SGD noise. Evci et al. (2022) find a pruning solution and mask using Gradual Magnitude Pruning (GMP; Zhu & Gupta (2017)) instead of IMP and apply the mask to the rewind point to obtain a lottery ticket solution. They then observe that these two sparse solutions are linearly mode connected, and close in function space. In contrast, our work studies linear mode connectivity between pairs of IMP solutions at *different* levels of sparsity. This allows us to understand the mechanism by which an IMP iteration can find a sparser matching subnetwork from the previous matching IMP solution. Finally, in concurrent work, Na et al. (2022) show that networks trained with SAM (Foret et al., 2021) are more amenable to pruning. This provides further evidence connecting loss landscape flatness and pruning. See Appendix A for extended related work.

## 2 PROBLEM SETUP, NOTATION, AND DEFINITIONS

Let $\mathbf{w} \in \mathbb{R}^D$ be the weights of a network and let $\mathcal{E}(\mathbf{w})$ be its test error on a classification task.

**Definition 2.1** (Linear Connectivity). *Two weights* $\mathbf{w}, \mathbf{w}'$ *are* $\varepsilon$-**linearly connected** *if* $\forall \gamma \in [0, 1]$,

$$\mathcal{E}(\gamma \cdot \mathbf{w} + (1 - \gamma) \cdot \mathbf{w}') \leq \gamma \cdot \mathcal{E}(\mathbf{w}) + (1 - \gamma) \cdot \mathcal{E}(\mathbf{w}') + \varepsilon. \quad (2.1)$$

*We define the* **error barrier** *between* $\mathbf{w}$ *and* $\mathbf{w}'$ *as the smallest* $\varepsilon$ *for which this is true. We say two weights are* **linearly mode connected** *if the error barrier between them is small (i.e., less than the standard deviation across training runs).*

**Sparse subnetworks.** Given a dense network with weights $\mathbf{w}$, a sparse subnetwork has weights $\mathbf{m} \odot \mathbf{w}$, where $\mathbf{m} \in \{0, 1\}^D$ is a ***binary mask*** and $\odot$ is the element-wise product. The ***sparsity*** of a mask $\mathbf{m}$ is the fraction of zeros, $(1 - \eta) \in [0, 1]$. Such a mask also defines an $\eta D$-dimensional ***axial subspace*** spanned by coordinate axis vectors associated with weights not zeroed out by the mask.

**Notation for training.** For an iteration $t$, weights $\mathbf{w}$, and number of steps $t'$, let $\mathcal{A}_t(\mathbf{w}, t')$ be the output of training the weights $\mathbf{w}$ for $t'$ steps starting with the algorithm state (e.g. learning rate schedule) at time $t$. In this notation, if $\mathbf{w}_0$ denotes the randomly initialized weights, then ordinary training for $T$ steps produces the final weights $\mathbf{w}_T = \mathcal{A}_0(\mathbf{w}_0, T)$.

**Iterative magnitude pruning (IMP).** IMP with Weight Rewinding (IMP-WR) is described in Algorithm 1 (Frankle et al., 2020a). Each pruning iteration is called a ***pruning level***, and $\mathbf{m}^{(L)}$ is the mask obtained after $L$ levels of pruning. $\tau$ denotes the ***rewind step***, $\mathbf{w}_\tau$ the ***rewind point***, and $1 - \alpha$ denotes a fixed pruning ratio, i.e. fraction of weights removed. The algorithm is depicted schematically in Fig. 1. The axial subspace associated with mask $\mathbf{m}^{(L)}$ is a colored subspace, the pruned rewind point $\mathbf{m}^{(L)} \odot \mathbf{w}_\tau$ is the circle in this subspace, the level $L$ solution $\mathbf{w}^{(L)}$ obtained from training is the triangle also in this subspace, and the level $L$ projection obtained as $\mathbf{m}^{(L+1)} \odot \mathbf{w}^{(L)}$ is the cross in the next, lower dimensional axial subspace. Note $\mathbf{w}^{(L)} = \mathcal{A}_\tau(\mathbf{m}^{(L)} \odot \mathbf{w}_\tau, T - \tau)$.

We also study two variants of IMP with different retraining strategies in Section 3.4: IMP with LR rewinding (IMP-LRR) and IMP with finetuning (IMP-FT) (Renda et al., 2020). In IMP-LRR, the level $L - 1$ solution, and not the rewind point, is used as the initialization for level $L$ retraining and the entire LR schedule is repeated: $\mathbf{w}^{(L)} = \mathcal{A}_0(\mathbf{m}^{(L)} \odot \mathbf{w}^{(L-1)}, T)$. IMP-FT is similar to IMP-LRR but instead of repeating the entire LR schedule, we continue training at the final low LR for the same number of steps: $\mathbf{w}^{(L)} = \mathcal{A}_T(\mathbf{m}^{(L)} \odot \mathbf{w}^{(L-1)}, T)$. See Appendix A for a discussion.

---

Algorithm 1: Iterative Magnitude Pruning-Weight Rewinding (IMP-WR) (Frankle et al., 2020a)

---

1: Initialize a dense network $\mathbf{w}_0 \in \mathbb{R}^d$ and a pruning mask $\mathbf{m}^{(0)} = 1^d$.
2: Train $\mathbf{w}_0$ for $\tau$ steps to $\mathbf{w}_\tau$.                                                            ▷ Phase 1: Pre-Training
3: **for** $L \in \{0, \dots, L_{\max} - 1\}$ **do**                                                              ▷ Phase 2: Mask Search
4:     Train the pruned network $\mathbf{m}^{(L)} \odot \mathbf{w}_\tau$ to obtain a level $L$ solution $\mathbf{w}^{(L)}$
5:     Prune a fraction $1 - \alpha$ of smallest magnitude nonzero weights after training:
       Let $\mathbf{m}^{(L+1)}[i] = 0$ if weight $i$ is pruned, otherwise let $\mathbf{m}^{(L+1)}[i] = \mathbf{m}^{(L)}[i]$.
6: Train the final network $\mathbf{m}^{(L_{\max})} \odot \mathbf{w}_\tau$. Measure its accuracy.                          ▷ Phase 3: Sparse Training

---

**Definition 2.2.** *A sparse network* $\mathbf{m} \odot \mathbf{w}$ *is* $\varepsilon$*-matching* **(in error)** *if it achieves accuracy within* $\varepsilon$ *of that of a trained dense network* $\mathbf{w}_T$: $\mathcal{E}(\mathbf{m} \odot \mathbf{w}) \leq \mathcal{E}(\mathbf{w}_T) + \varepsilon$.

Since we always set $\varepsilon$ as the standard deviation of the error of independently trained dense networks, we drop the $\varepsilon$ from our notation and simply use *matching*.

**Definition 2.3.** *A (sparse or dense) network* $\mathbf{w}_\tau$ *is* **stable** *(at iteration* $\tau$*) if, with high probability, training a pair of networks from initialization* $\mathbf{w}_\tau$ *using the same algorithm but with different randomness (e.g., different minibatch order, data augmentation, random seed, etc.) produces final weights that are linearly mode connected. The first iteration* $\tau$ *at which a network is stable is called the* **onset of linear mode connectivity (LMC)***.*

Frankle et al. (2020a) provide evidence that sparse subnetworks are matching if and only if they are stable. Finally, playing a central role in our work is the concept of an LCS-set.

**Definition 2.4.** *An* $\varepsilon$*-linearly connected sublevel set (LCS-set) of a network* $\mathbf{w}$ *is the set of all weights* $\mathbf{w}'$ *that achieve the same error as* $\mathbf{w}$ *up to* $\varepsilon$, *i.e.* $\mathcal{E}(\mathbf{w}') \leq \mathcal{E}(\mathbf{w}) + \varepsilon$, *and are linearly mode connected to* $\mathbf{w}$, *i.e. there are no error barriers on the line connecting* $\mathbf{w}$ *and* $\mathbf{w}'$ *in weight space.*

Note that an LCS-set is star convex; not all pairs of points in the set are linearly mode connected. As with matching, we drop $\varepsilon$ from the notation for LCS-set.

## 3 RESULTS

### 3.1 PRUNING MASKS IDENTIFY AXIAL SUBSPACES THAT INTERSECT MATCHING LCS-SETS.

First, we elucidate what useful information the mask found at the end of training at level $L$ provides to the rewind point at level $L + 1$. We find that when an iteration of IMP from level $L$ to $L + 1$ finds

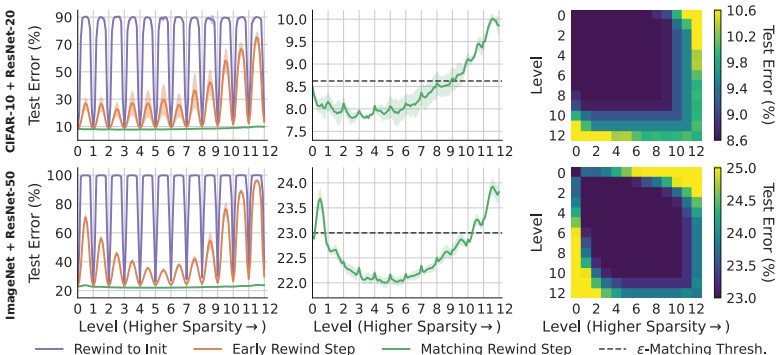

Figure 2: Error Connectivity of IMP. **Left:** The error along a piecewise linear path connecting each level $L$ solution $\mathbf{w}^{(L)}$ to the $L+1$ solution $\mathbf{w}^{(L+1)}$ for 3 different rewind steps. Purple curves: rewind step 0, orange curves: early rewind steps (250 for CIFAR-10 and 1250 for ImageNet), and green curves: rewind steps that produce matching subnetworks (2000 for CIFAR-10 and 5000 for ImageNet). **Middle:** Zoomed in plot on green curve shows that networks on the piecewise linear path between matching IMP solutions are also matching. **Right column:** The maximal error barrier between all pairs of solutions at different levels (from the matching rewind step). The dark blue regions indicate solutions in a matching linearly connected mode. See Fig. 16 for CIFAR-100/ResNet-18 results and Fig. 11 for results on Iterative Gradient Pruning, a different pruning strategy.

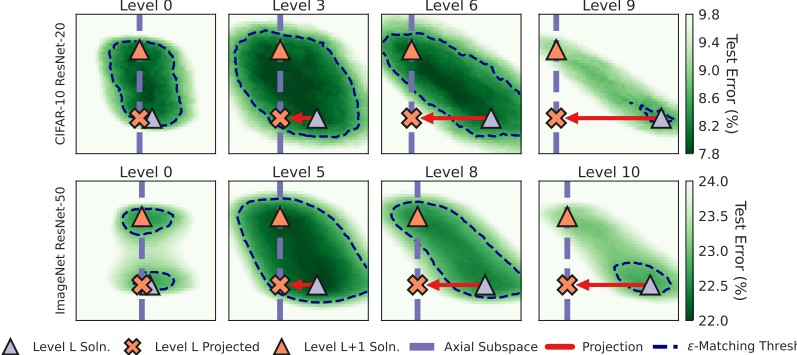

Figure 3: Two-dimensional slices of the error landscape spanned by 3 points at each level $L$: **(1)** the solution $\mathbf{w}^{(L)}$ (grey triangle), **(2)** its level $L$ projection $\mathbf{m}^{(L+1)} \odot \mathbf{w}^{(L)}$ (orange $\times$) onto the axial subspace $\mathbf{m}^{(L+1)}$ (purple dashed line), and **(3)** the level $L+1$ solution $\mathbf{w}^{(L+1)}$ (orange triangle) found by retraining with the new mask $\mathbf{m}^{(L+1)}$. The axial subspace $\mathbf{m}^{(L+1)}$ is obtained by 20% magnitude pruning on $\mathbf{w}^{(L)}$. The dotted black contour outlines the LCS-set of $\mathbf{w}^{(L)}$. Column 2 shows a low sparsity level where the projection remains within the LCS-set; column 3 shows a higher sparsity level where the projection is outside, but $\mathbf{w}^{(L+1)}$ returns to the LCS-set. Column 4 shows a higher sparsity level, at which IMP fails to find a matching solution: both the projection and retrained solution lie outside the LCS-set. See Fig. 17 and 18 for additional results.

a matching subnetwork, the axial subspace $\mathbf{m}^{(L+1)}$ obtained by pruning the level $L$ solution, $\mathbf{w}^{(L)}$, intersects the LCS-set of this solution. By the definition of LCS-set, all the points in this intersection are matching solutions in the sparser $\mathbf{m}^{(L+1)}$ subspace and are linearly connected to $\mathbf{w}^{(L)}$. We also find that the network $\mathbf{w}^{(L+1)}$ found by SGD is in fact one of these solutions. Conversely, when IMP from level $L$ to $L+1$ does not find a matching subnetwork, the solution $\mathbf{w}^{(L+1)}$ does not lie in the LCS-set of $\mathbf{w}^{(L)}$, suggesting that the axial subspace $\mathbf{m}^{(L+1)}$ does not intersect this set. Thus, we hypothesize that a round of IMP finds a matching subnetwork if and only if the sparse axial subspace found by pruning intersects the LCS-set of the current matching solution.

Figs. 2 and 3 present evidence for this hypothesis. The left and center columns of Fig. 2 show that in a ResNet-50 (ResNet-20) trained on ImageNet (CIFAR-10), for rewind steps at initialization (blue curve) or early in training (orange curve), successive IMP solutions $\mathbf{w}^{(L)}$ and $\mathbf{w}^{(L+1)}$ are *neither* matching *nor* linearly mode connected. However, at a later rewind point (green curve) successive

matching solutions are linearly mode connected. Fig. 3 visualizes two dimensional slices of the error landscape containing the level $L$ solution, its pruning projection, and the level $L + 1$ solution. We find that at early pruning levels, the projected network, $\mathbf{m}^{(L+1)} \odot \mathbf{w}^{(L)}$, remains in the LCS-set of $\mathbf{w}^{(L)}$. Thus the $\mathbf{m}^{(L+1)}$ axial subspace intersects this set. As $L$ increases, the projections leave the LCS-set of $\mathbf{w}^{(L)}$, which also shrinks in size. However, the axial subspace $\mathbf{m}^{(L+1)}$ still intersects the LCS-set of $\mathbf{w}^{(L)}$ since $\mathbf{w}^{(L+1)}$ lies in this set. Conversely, at the sparsity level when matching breaks down, the axial subspace no longer intersects the LCS-set.

In summary, when IMP succeeds, i.e. $\mathbf{w}^{(L)}$ and $\mathbf{w}^{(L+1)}$ are both matching, the mask $\mathbf{m}^{(L+1)}$ defines an axial subspace that intersects the LCS-set of $\mathbf{w}^{(L)}$. When IMP fails, this intersection also fails. Thus, the key information provided by the mask $\mathbf{m}^{(L+1)}$ is a good axial subspace that could potentially guide SGD to matching solutions in the LCS-set of $\mathbf{w}^{(L)}$ but at a higher sparsity.

Note that at rewind steps where IMP is successful, Level 0 and Level 1 may not be linearly mode connected (Fig. 3 bottom left) because the dense network is not yet stable (Frankle et al., 2020a). However, we sill find matching solutions as the network with 80% weights remaining is still very overparameterized and many good optima exist. See Appendix G for a detailed discussion.

Another interesting observation: the dark blue regions in Fig. 2 (right) indicate that all pairs of matching IMP solutions at intermediate levels are linearly mode connected with each other. However, in ImageNet, there are error barriers between the earliest and last matching level (yellow block at position (1, 10)). Though each successive pair of matching IMP solutions are linearly connected, *all* matching IMP solutions need not lie in a convex linearly connected mode. The connected set containing the piecewise linear path between successive IMP solutions can in fact be quite non-convex; see Fig. 16 for an extreme example on CIFAR-100/ResNet-18. Also see Appendix H for results on Iterative Gradient pruning, another iterative pruning method.

## 3.2 Retraining finds matching subnetworks if SGD is robust to perturbations.

So far we have seen that IMP succeeds in finding a matching subnetwork if $\mathbf{w}^{(L)}$ is matching and the mask $\mathbf{m}^{(L+1)}$ provides an axial subspace that intersects the LCS-set of $\mathbf{w}^{(L)}$, thus guaranteeing the existence of sparse matching solutions. But how is SGD able to extract this information from the mask at the rewind point. In particular, assuming $\mathbf{m}^{(L+1)}$ intersects the LCS-set of $\mathbf{w}^{(L)}$, why would $\mathbf{m}^{(L+1)} \odot \mathbf{w}_\tau$ train to a $\mathbf{w}^{(L+1)}$ that lies within the LCS-set of $\mathbf{w}^{(L)}$ and not some other optimum in the $\mathbf{m}^{(L+1)}$ axial subspace?

We will show that IMP retraining likely does so because of the robustness of SGD training to perturbations at late enough rewind steps. We treat $\mathbf{v} = \left(\mathbf{m}^{(L+1)} \odot \mathbf{w}_\tau\right) - \left(\mathbf{m}^{(L)} \odot \mathbf{w}_\tau\right)$ as a perturbation to the level $L$ pruned rewind point $\mathbf{m}^{(L)} \odot \mathbf{w}_\tau$. At this rewind point, we hypothesize that the network $\mathbf{m}^{(L)} \odot \mathbf{w}_\tau$ is stable not only up to SGD noise (Frankle et al., 2020a), but also that the linearly connected mode SGD trains to is robust to perturbations of magnitude less than or equal to $\|\mathbf{v}\|_2$. We define robustness of SGD to perturbations as follows:

**Definition 3.1.** *A (sparse) network $\mathbf{w}_\tau$ is **robust at $\tau$ to random perturbations of size** $c$, if with high probability $\mathcal{A}_\tau(\mathbf{w}_\tau, T - \tau)$ is linearly connected to $\mathcal{A}_\tau(\mathbf{w}_\tau + \mathbf{v}, T - \tau)$, for $\mathbf{v}$ drawn uniformly from the sphere of radius $c$ around $\mathbf{0}$, $S_c(\mathbf{0})$.*

Intuitively, this means that weights a distance $c$ away from $\mathbf{w}_\tau$ are in the same attraction basin as $\mathbf{w}_\tau$. If SGD is robust to perturbations of the same norm as $\|\mathbf{v}\|_2$, then $\mathbf{m}^{(L+1)} \odot \mathbf{w}_\tau$ will train to a solution that is linearly mode connected to $\mathbf{w}^{(L)}$. Additionally if $\mathbf{w}^{(L)}$ is matching and $\mathbf{m}^{(L+1)}$ intersects the LCS-set of $\mathbf{w}^{(L)}$, then $\mathbf{w}^{(L+1)}$ is in the LCS-set of $\mathbf{w}^{(L)}$ and so it will be matching. We empirically test this notion of SGD robustness starting from the pruned rewind point $\mathbf{m}^{(L)} \odot \mathbf{w}_\tau$ for all levels $L$ and perturb by either the actual IMP perturbation $\mathbf{v}$ or a random perturbation of norm $\|\mathbf{v}\|_2$. Fig. 4 confirms that whenever IMP can find a matching network, SGD is indeed robust not only to IMP-induced perturbations but also random perturbations. In Appendix D, we show implications of this result: masks found by IMP-LRR can also be used for retraining from the rewind point. In summary, we find that IMP can use the mask $\mathbf{m}^{(L+1)}$ at an early rewind point to navigate back to an LCS-set of the previous solution $\mathbf{w}^{(L)}$ when the rewind point enjoys the property of SGD robustness to sufficiently large perturbations. See Appendix I for further experiments on robustness of SGD.

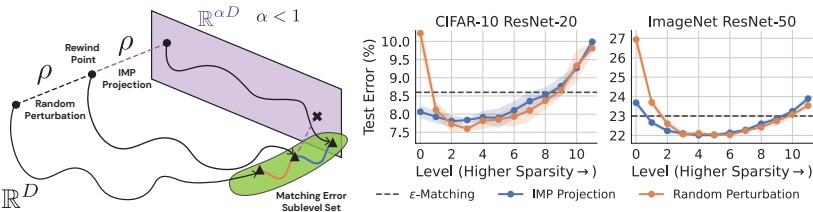

Figure 4: Robustness of SGD. **Left:** At a rewind step $\tau$ that yields a winning ticket $\mathbf{w}_\tau$, for each level $L$, we add a random perturbation to the level $L$ pruned rewind point $\mathbf{m}^{(L)} \odot \mathbf{w}_\tau$ with the same $l_2$ norm as the $l_2$ distance $R$ between $\mathbf{m}^{(L)} \odot \mathbf{w}_\tau$ and level $L+1$ pruned IMP projection $\mathbf{m}^{(L+1)} \odot \mathbf{w}_\tau$. We then train all 3 networks (black curves) **Right:** The error barrier between the level $L$ solution $\mathbf{w}^{(L)}$ and the level $L+1$ IMP solution $\mathbf{w}^{(L+1)}$ (blue) and the error barrier between the level $L$ solution $\mathbf{w}^{(L)}$ and the trained random perturbation (orange). These curves show that for levels that yield matching solutions, the robustness of SGD explains why, after projecting the rewind step, $\mathbf{m}^{(L+1)} \odot \mathbf{w}_\tau$ can train to the LCS-set of $\mathbf{w}^{(L)}$. See Fig 19 for CIFAR-100/ResNet-18 results.

### 3.3 THE HESSIAN EIGENSPECTRUM GOVERNS MAXIMAL PRUNING RATIOS PER ITERATION.

Our third question concerns why iterative pruning is necessary and why we can't simply prune to high sparsities in one-shot without sacrificing accuracy? To address this, consider the level $L$ IMP solution $\mathbf{w}^{(L)}$ and its associated projection $\mathbf{m}^{(L+1)} \odot \mathbf{w}^{(L)}$ (i.e. triangles and their associated $\times$'s in Fig. 1). Let us denote the distance between $\mathbf{m}^{(L+1)} \odot \mathbf{w}^{(L)}$ and $\mathbf{w}^{(L)}$ by $R$. Also we know from Sec. 3.1 that when IMP achieves matching, the axial subspace $\mathbf{m}^{(L+1)}$ intersects the LCS-set of $\mathbf{w}^{(L)}$ because retraining within $\mathbf{m}^{(L+1)}$ yields a matching $\mathbf{w}^{(L+1)}$ that is linearly connected to $\mathbf{w}^{(L)}$. Now suppose we have used a constant pruning ratio (fraction of weights removed) of $1 - \alpha < 1$ up to level $L$ so that $\mathbf{w}^{(L)}$ has $\alpha^L D$ nonzero weights. Suppose further that at level $L$ we explore different pruning ratios $f$. This means we remove a fraction $f$ of the $\alpha^L D$ nonzero weights in $\mathbf{w}^{(L)}$ to obtain networks in an axial subspace $\mathbf{m}^{(L+1)}$ that have $(1 - f)\alpha^L D$ nonzero weights. Thus we would like to understand conditions under which an axial subspace of dimension $d = (1 - f)\alpha^L D$ centered at a point $\mathbf{m}^{(L+1)} \odot \mathbf{w}^{(L)}$ of distance $R$ from $\mathbf{w}^{(L)}$ intersects the LCS-set of this solution. This leads us to exploit the phase transition observed in Larsen et al. (2022):

**Phase transitions in the intersection probability of random subspaces with error sublevel sets.** While it is difficult to characterize when an *axial* subspace of low dimension intersects a given error sublevel set, recent work has analyzed both empirically (Li et al., 2018) and theoretically (Larsen et al., 2022) when a *random* affine subspace of dimension $d$ centered at a point $P$ intersects a loss sublevel set located a distance $R$ from $P$. In particular Larsen et al. (2022) showed that the intersection probability (over the choice of random affine subspaces) undergoes a sharp phase transition from 0 to 1 as the dimensionality $d$ increases. We call the dimensionality $d^*$ at which this phase transition occurs the intersection threshold (it was called the threshold training dimension by Larsen et al. (2022)). This threshold increases the further the point $P$ is from the sublevel set (it is harder to hit a sublevel set when one starts further away) and decreases the larger the sublevel set is (it is easier to hit larger sublevel sets). Larsen et al. (2022) found general formulas for the intersection threshold $d^*$, and derived an explicit formula when the error landscape could be approximated as a quadratic well, yielding:

**Lemma 3.1.** *Consider a quadratic error landscape over an ambient dimension $D^A$ with minimum error $0$ and Hessian eigenvalues $\{\lambda_1, \ldots, \lambda_{D^A}\}$. An $\varepsilon$-sublevel set $M^\varepsilon$ is then an ellipsoid with principal radii $r_i = \sqrt{2\varepsilon/\lambda_i}$. A random affine subspace of dimension of $d$ centered at a point $P$ a distance $R$ from the minimum intersects $M^\varepsilon$ with high (low) probability if $d > d^*$ ($d < d^*$) where the intersection threshold $d^*$ is given by the approximate upper bound $d^* = D^A - \sum_i r_i^2/(R^2 + r_i^2)$.*

Intuitively, the further one is from the minimum (larger $R$), the higher the intersection threshold is (harder to hit). Also the larger the sublevel set is (through more large radii $r_i$ corresponding to a Hessian eigenspectra with more small $\lambda_i$, i.e. flat directions) the lower the intersection threshold $d^*$ (i.e. the easier it is to intersect). See Appendix C for more discussion.

**The intersection thresholds of random subspaces predicts maximal pruning ratios.** We now apply Lemma 3.1 to predict maximal pruning ratios for random affine subspaces, which we use as

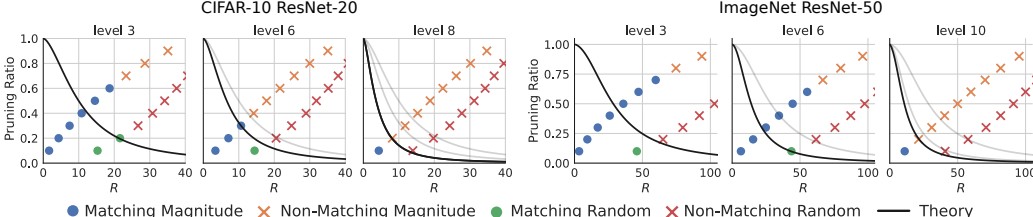

Figure 5: The Hessian eigenspectrum determines the limits of IMP. For each level $L$, we use the intersection threshold of random subspaces with LCS-sets (see Lemma 3.1 and main text) to derive a maximal theoretical pruning ratio at level $L$ (black curves) that still allows matching. This predicted maximal ratio decreases with the distance $R$ of the projection. The resultant limiting black curve shrinks to the origin as the Hessian eigenspectrum gets sharper at higher levels $L$. We illustrate this by plotting the theoretically predicted maximal pruning ratio curves from earlier levels in light gray. We compare the maximum pruning ratio with both random and magnitude pruning. The theoretical maximum pruning ratio (black curves) predicts well when random pruning will match at small pruning ratios (green circles below black curves) and when random pruning will fail to match at large pruning ratios (red $\times$'s above black curves). Interestingly, IMP slightly outperforms random pruning, with some matching IMP solutions (blue circles) occuring above the black curves, though eventually they fail to match at even larger pruning ratios (orange $\times$'s).

an approximation for pruning to a random axial subspace. In particular, we consider $\mathbf{w}^{(L)}$ to be at the minimum of an error landscape over ambient dimension $D^A = \alpha^L D$. We approximate this error landscape through a quadratic approximation, using the Lanczos algorithm (Yao et al., 2020) to estimate the *entire* Hessian eigenspectrum density at $\mathbf{w}^{(L)}$. We then prune $\mathbf{w}^{(L)}$ with pruning ratio $f$ (either random pruning or magnitude pruning) obtaining an affine axial subspace of dimension $d = (1 - f)D^A$ centered at point $P = \mathbf{m}^{(L+1)} \odot \mathbf{w}^{(L)}$ that is a distance $R$ from $\mathbf{w}^{(L)}$. We model this axial subspace as a random affine subspace. Then inserting these expressions into the intersection phase transition condition $d > d^* = D^A - \sum_i r_i^2/(R^2 + r_i^2)$ turns a lower bound on the subspace dimension $d = (1 - f)D^A$ in order to intersect the sublevel set into an upper bound on the pruning ratio $f$ in order to intersect. Thus, according to this random subspace intersection phase transition, the intersection of the axial subspace $\mathbf{m}^{(L+1)}$ with the LCS-set of $\mathbf{w}^{(L)}$ is predicted to occur *only* if the pruning ratio $f$ is less than a threshold set by both the distance $R$ and the radii $r_i$. In particular, smaller distances $R$ moved in the projection allow larger pruning ratios $f$, as do Hessian eigenspectra with more flat directions, i.e. with more large radii $r_i$.

Fig. 5 shows the results of comparing the prediction of Lemma 3.1 to both random and magnitude pruning at a variety of pruning ratios $f$ and levels $L$. Our theory predicts well when random pruning shifts from matching—at small pruning ratios $f$ below the predicted maximum—to non-matching—at large pruning ratios above the maximum. Furthermore, as one progresses to higher sparsity levels $L$, the maximal allowed pruning ratio (black phase boundaries) shift lower. This happens because the error landscape is getting sharper at higher sparsity levels, thereby limiting the maximal allowed pruning ratio more stringently.

**IMP preferentially prunes flatter landscape directions.** Surprisingly, we also note in Fig. 5 that IMP can sometimes find matching subnetworks at pruning ratios higher than the maximal allowed by our theory given the assumptions—a few blue circles to the upper right of the black curves. This suggests that the success of IMP is not due to a smaller projection distance $R$ alone. Strikingly, the small magnitude weights pruned by IMP are preferentially correlated with flatter error directions in the LCS-set. Fig. 6 provides strong evidence for this hypothesis. IMP implicitly pruning in a flatter direction of the error landscape allows for higher pruning ratios. We note that pruning weights along flat directions dates back to the 1980's, e.g. optimal brain damage (LeCun et al., 1989).

## 3.4 THE IMPORTANCE OF ITERATIVE PRUNING AND RETRAINING

Here we study our fourth question: the role of retraining. Renda et al. (2020) find that both IMP-WR and IMP-LRR find matching subnetworks while standard finetuning (with fixed small learning rates) does not. What does either weight or LR rewinding achieve that typical finetuning does not? Fig. 7 demonstrates that both IMP-WR and IMP-LRR reequilibrate the weight distribution after pruning, i.e. a retrained network once again contains a substantial fraction of small magnitude weights that

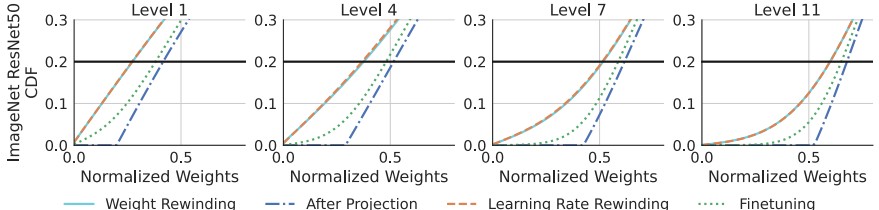

Figure 6: **Magnitude pruning directions correlate with flat directions.** The level $L$ solution $\mathbf{w}^{(L)}$ is at the center. The orange (purple) arrow indicates weight space motion when we prune the 20% smallest (random) weights. We visualize a slice of the error landscape along the plane spanned by these two directions and the black dashed contour shows the boundary of the LCS-set of $\mathbf{w}^{(L)}$. Note: (1) the smallest magnitude weight direction is flatter than the random one; and (2) the basin in this slice becomes sharper at higher levels of sparsity. See Fig. 20 and 21 for CIFAR-10 and CIFAR-100.

Figure 7: The effect of Weight Rewinding, Learning Rate Rewinding and Finetuning on the distribution of weights at each level. The dark blue curve shows the cumulative distribution function (CDF) of the absolute value of the normalized weights immediately after projection; the initial flat segment is due to the 20% of the weights pruned from the previous level. The three other curves show the CDF of the weights after three different training procedures: IMP-WR, IMP-LRR, and IMP-FT. We observe that IMP-WR and IMP-LRR produce similar CDFs but IMP-FT results in fewer small weights. These observations indicate that a key role of iteration in IMP is to reequilibriate the weights. See Fig. 9 for a discussion and Fig. 22 for CIFAR-10 results.

are amenable to pruning. Finetuning, on the other hand, fails to reequilibrate the weights and further pruning creates large projections, thus making it difficult to find an axial subspace that intersects with the LCS-set containing the current solution. As a result, finetuning fails to find matching networks up to the same levels of sparsity as weight or learning rate rewinding. Further discussion in Fig. 9.

# 4 DISCUSSION

In this work, we investigate the different steps that make up an iteration of IMP and construct a scientific understanding of the role played by each of these steps in finding winning lottery tickets. We uncover that the IMP mask conveys to the rewind point the location of a linearly connected mode containing matching sparse solutions. We show that the ability to train into this mode from the rewind point and find these matching solutions follows from the robustness of SGD at the rewind step. We also forge a new link between the Hessian eigenspectrum and IMP, showing sharper minima limit maximal pruning ratios. Remarkably, we discover that IMP implicitly finds flatter directions to prune. Finally, we show that the role of retraining is to find new small-magnitude weights to prune.

Our results suggest that a key design principle for future pruning strategies might involve direct exploration of lower-dimensional axial subspaces that intersect the LCS-set of the current network. Indeed, motivated by our new scientific understanding we show in Appendix F (Fig. 10) that it is possible to achieve the same performance as IMP but with fewer levels of pruning by dynamically choosing the per-level pruning ratio so as to stay within the LCS-set after projection. In a CIFAR-100 example, we are able to prune to the same sparsity in 7 levels instead of the 11 required when using the fixed ratio of 20%. Though the strategy of pruning more for low sparsity levels is not new, our findings provide new geometric principles for efficiently and effectively choosing this pruning ratio hyperparameter at every sparsity level. Additionally, by uncovering that the key role of retraining is to reequilibrate the weights, our results provide a new direction for algorithmic approaches to achieve this property using less compute than full retraining. Finally, our work opens up new theoretical questions, such as why small magnitude weights correlate with flatter directions on the error landscape.

ACKNOWLEDGMENTS

The experiments for this paper were partially funded by Google Cloud research credits and partially performed at Meta AI. The authors would like to thank Daniel M. Roy, Ari Morcos, and Utku Evci for feedback on drafts.

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

## A    EXTENDED RELATED WORK

**Rewind point.**    Frankle et al. (2020a) observed that, for larger datasets and architectures, IMP fails to find matching subnetworks at random initialization. However, matching subnetworks can instead be found after briefly training the dense network and using these new pre-trained weights as the rewind point. The authors further observed that the rewind step at which matching subnetworks emerge strongly correlates with the onset of linear mode connectivity in the pruned network. They hypothesize that IMP is able to find matching initializations once SGD has "stabilized" in the sparse network's subspace. Follow up work by Paul et al. (2022) characterized the role of data in this pre-training stage, and analyzed what information is encoded into this rewind point by the dense network training. The rewind step alone, however, cannot explain IMP's success because random masks applied to this point do not produce matching lottery ticket initializations (Frankle et al., 2020a). Therefore, in addition to the information encoded in the rewind point, there must be some critical information encoded *in the mask* itself.

**Masks constructed at the end of training.**    Significant attention has focused on trying to find matching subnetworks early in training or at initialization without information after convergence (e.g., Lee et al., 2019; Wang et al., 2020; Tanaka et al., 2020). Despite a lot of progress, these early pruning methods fall short of finding matching initializations (Su et al., 2020; Frankle et al., 2021). As far as we know, a key component of the success of IMP remains the use of information from the end of training to construct the mask (e.g., Zhou et al., 2019; Frankle et al., 2020a; Ramanujan et al., 2020; Savarese et al., 2020; Sreenivasan et al., 2022).

Our work identifies the mechanism by which IMP finds matching solutions: the algorithm maintains the information from the dense network about the loss landscape by encoding this information into the mask. Evci et al. (2022) report similar findings but for a different setting. In their work, they construct sparse masks from a pruned solution (sparse network trained to convergence), where the latter is obtained through gradual magnitude pruning (GMP) throughout training (Zhu & Gupta, 2017) as opposed to IMP. They find that the resulting sparse subnetwork lies in the same basin (i.e., no error barrier on a connecting linear path) as the original pruned solution. Note that the GMP pruned solution does not match the accuracy obtained by the dense solution, which is one of the key properties of sparse subnetworks studied in our work. These results hint at the larger picture explored in this paper about the full sequence of matching networks of increasing sparsity found by IMP being piecewise linearly connected in the error landscape.

**Iterative pruning with retraining.**    An essential part of IMP is that the weights are pruned iteratively with periods of retraining between them. For standard training without rewinding, Han et al. (2015) show that one-shot magnitude pruning cannot find subnetworks of the same sparsity as iterative magnitude pruning; Frankle & Carbin (2018) show the same for the lottery ticket setting in which weights are rewound to their values from early in training after pruning. Alternative methods that attain matching performance at the sparsity levels as IMP also feature iterative pruning and retraining (Renda et al., 2020; Savarese et al., 2020). In this work, we take the first step towards understanding why the iterative piece of IMP is critical for finding high sparsity and yet matching subnetworks.

**Finetuning and learning rate rewinding.**    In the IMP-WR framework proposed by Frankle et al. (2020a) after each pruning step the network is rewound to an early rewind point $\mathbf{w}_\tau$, and from that point on the network is retrained with the new sparsity pattern. One can alternatively consider methods that train from the final model after pruning rather than this rewind point. We refer to IMP-FT as training the pruned final model with the same final learning rate. Renda et al. (2020) show that this finetuning underperforms IMP-WR in final test accuracy. To alleviate this, Renda et al. (2020) propose a middle ground between IMP-WR and IMP-FT called learning rate rewinding (IMP-LRR). In IMP-LRR, the final pruned model is trained for the same amount of time as the original training run and with the original learning rate schedule, but starting at the weights at convergence instead of rewinding to $\mathbf{w}_\tau$. As shown in Renda et al. (2020), this produces networks that perform equivalently to IMP-WR.

**Pruning flat directions.**    The basic idea of pruning the weights that are aligned with "flat" directions in the optimization landscape dates back to late 1980's, and was explicitly described in the

work introducing optimal brain damage (LeCun et al., 1989). The motivation is based on a second order Taylor expansion of the optimization objective $g(\mathbf{w}, S)$. In more detail, for an axis-aligned perturbation captured by a mask $\mathbf{m}$,

$$g(\mathbf{m} \odot \mathbf{w}, S) - g(\mathbf{w}, S) \approx ((1 - \mathbf{m}) \odot \mathbf{w})^T \frac{\mathrm{d}g(\mathbf{w}, S)}{\mathrm{d}\mathbf{w}} + ((1 - \mathbf{m}) \odot \mathbf{w})^T \mathbf{H}((1 - \mathbf{m}) \odot \mathbf{w}), \quad \text{(A.1)}$$

where $\mathbf{H} = \mathrm{d}^2 g(\mathbf{w}, S)/\mathrm{d}\mathbf{w}^2$. This requires computing all Hessian entries $(i, j)$ for which $(\mathbf{m})_i, (\mathbf{m})_j = 0$. Finding masks that minimize the change in the optimization objective $g(\mathbf{w}, S)$ requires computing the full spectrum of the Hessian and identifying the smallest eigenvalue directions. This is prohibitively expensive in modern deep neural networks. Last but not least, ideally we want to look at the empirical error or error landscapes instead, which are not differentiable.

## B  EXPERIMENTAL DETAILS

**CIFAR-10 ResNet-20.**  We train with SGD and a batchsize of 128 for 62400 steps. We use lr = 0.1, momentum = 0.9, weight decay = 0.0001. The learning rate is decayed by a factor or 10 at 31200 and 46800 steps. We run 12 rounds of pruning and prune 20% of the smallest magnitude weights at each round. After each round of pruning, the weights are rewinded to 0, 250, or 2000 steps and then retrained with the sparsity mask corresponding to that level. For each rewind step we plot the mean and standard deviation of the final test accuracies across 4 replicates with independent random seeds.

**CIFAR-100 ResNet-18.**  We train with SGD and a batchsize of 128 for 78125 steps. We use lr = 0.1, momentum = 0.9, weight decay = 0.0005. The learning rate is decayed by a factor or 5 at 23438, 46875, and 62500 steps. We run 15 rounds of pruning and prune 20% of the smallest magnitude weights at each round. After each round of pruning, the weights are rewinded to 0, 400, or 3200 steps and then retrained with the sparsity mask corresponding to that level. For each rewind step we plot the mean and standard deviation of the final test accuracies across 4 replicates with independent random seeds.

**ImageNet ResNet-50.**  We train with decoupled SGD (Loshchilov & Hutter, 2018) and a batchsize of 2048 for 15970 steps. We use lr = 2.048, momentum = 0.875, weight decay = 0.0005. We use cosine decay for learning rate scheduler with a warm-up of 5000 steps. We use additional algorithms to speed up the training as described by Leavitt (2022). We run 12 rounds of pruning and prune 20% of the smallest magnitude weights at each round. Additionally, After each round of pruning, the weights are rewinded to 0, 1250, or 5000 steps and then retrained with the sparsity mask corresponding to that level. For each rewind step we plot the mean and standard deviation of the final test accuracies across 4 replicates with independent random seeds.

**Pruning.**  Following Frankle & Carbin (2018), in all our pruning experiments, the prunable parameters are weights of the convolutional layers and the fully-connected layers. For experiments involving interpolation, we interpolate all parameters (both prunable and nonprunable parameters) within the convex hull of the models. When extrapolating outside the convex hall, we only extrapolate the prunable parameters and the non-prunable parameters are projected onto the closest boundary of the convex hull.

**Error Connectivity of IMP Solutions. (Fig. 2)**  In Fig. 2, we consider the error along linear paths between pairs of solutions found by IMP. The solution at level L is obtained after the Lth iteration of Algorithm 1 for a given rewind step. Given two IMP solutions, $\mathbf{w}^{(L)}$ and $\mathbf{w}^{(K)}$, we calculate the error along the linear interpolation of the two solutions: $\mathcal{E}((1 - \beta)\mathbf{w}^{(L)} + \beta\mathbf{w}^{(K)})$, where $\beta \in [0, 1]$. Typically, we evaluate beta at $\{0.1, 0.2, ..., 0.9\}$. We plot the test error along this path between IMP solutions $\mathbf{w}^{(L)}$ and $\mathbf{w}^{(L+1)}$. For all results we show the mean and standard deviation of 4 independent runs.

For the heatmap, between each pair of solutions given by row $i$ and column $j$, we calculate $\max_\beta \mathcal{E}((1 - \beta)\mathbf{w}^{(i)} + \beta\mathbf{w}^{(j)})$. This is also the mean of 4 runs. The darkest limit of the colorbar is set to the $\varepsilon$-matching threshold.

**Error landscape of an IMP step. (Fig. 3)** To investigate the error landscape of an IMP step, we evaluate the test error on the two-dimensional plane spanned by 3 points at each level $L$: the level $L$ solution $\mathbf{w}^{(L)}$, its level $L$ projection $\mathbf{m}^{(L+1)} \odot \mathbf{w}^{(L)}$ and the level $L+1$ solution $\mathbf{w}^{(L+1)}$. We construct a $51 \times 51$ grid and evaluate the error on the the grid. The contour is plotted for the matching error at each level for every dataset. For better visualization, we use different scales for the vertical and horizontal direction at each level. We fix the the positions of the level $L$ projection $\mathbf{m}^{(L+1)} \odot \mathbf{w}^{(L)}$ and the level $L+1$ solution $\mathbf{w}^{(L+1)}$ on each plot and scale the orthogonal direction by 2 times. We show the error landscape for all the levels in Fig. 17 to 19.

**Robustness at rewind point. (Fig. 4)** In this experiment, we compare the error barriers between two IMP solutions and an IMP solution and the trained solution after a random perturbation. For each level, the blue lines are the error barrier between the IMP solution at Level L and the IMP solution at Level L+1. These are just the midpoints between the successive level interpolations in Fig. 2. To obtain the orange points, we first calculate the distance of the projection when the magnitude pruning mask obtained by pruning the Level L IMP solution is applied to the rewind step. In the conceptual figure accompanying Fig. 4, this is represented by $\rho$. We then apply a random perturbation to the rewind step in the full $\alpha^L D$ dimensions of the Level L solution and train to convergence. The orange points are the test error halfway along the line connecting this solution and the original Level L IMP solution. All results show the mean and standard deviation of 4 independent runs.

**Threshold training dimension. (Fig. 5)** We first estimate an appropriate $\varepsilon_{\text{train}}^{(L)}$ for a linearly connected *training loss* sublevel set for level $L$. We perform random pruning at level 4 and record the train loss and test error. We repeat the procedure and linearly fit loss versus error to get the train loss $\mathcal{L}_{\text{matching}}$ corresponding to the test error $\mathcal{E}_{\text{dense}}$ for the dense network. We then determine $\varepsilon_{\text{train}}^{(L)}$ by subtracting the train loss of level $L$ solution from $\mathcal{L}_{\text{matching}}$. At each level, we use the PyHessian package (Yao et al., 2020) to estimate the Hessian eigenspectrum density of the train loss (average across 4 runs of 512 Lanczos iterations for CIFAR-10 and 2 runs of 128 Lanczos iterations for ImageNet; we randomly sample 25000 examples from ImageNet to evaluate the train loss for each run). In Fig. 5, the theory lines are generated from Lemma 3.1 with the estimated Hessian eigenspectrum. To verify the theory, we perform a series of pruning experiments at each level $L$. We scan the pruning ratio from $0.1$ to $0.9$ with a step of $0.1$ for both magnitude pruning and random pruning and check if the $L+1$ solution is matching. The projection distance from pruning is calculated as $R = \|w^{(L+1)} - w^{(L)}\|_2$.

**Small weights are correlated with flat directions. (Fig. 6)** We find that magnitude pruning outperforms the theoretical prediction. To investigate the reason, we study the properties of the LCS-set. In Fig. 6 and 20), we visualize the error landscape in the subspace spanned by a random projection and the magnitude projection (both for a 20% pruning ratio). We construct a $51 \times 51$ grid and evaluate the error on the the grid. The contour is plotted for the matching error at each level for every dataset. We use the same scale for both directions and for each level. Generally, the random pruning projection is not orthogonal to the the magnitude pruning projection and we have performed Gram-Schmidt procedure to figure out an orthogonal basis.

**The importance of iterative pruning and retraining. (Fig. 7)** In Fig. 7, we plot the CDF of weights at level $L$ projection $\mathbf{m}^{(L)} \odot \mathbf{w}^{(L-1)}$. We perform three different training procedures at level $L$: retraining from the rewind point (weight rewinding), FineTuning and Learning rate Rewinding and we plot the CDFs of the solution obtained from these three training procedures. When we plot the CDF, we normalize the absolute values of the weights by their means. For Learning Rate Rewinding, we rewind the original learning rate schedule and run the training for the same amount of time as the original run. For Fine Tuning, we use a learning rate of 0.02, which is 100 times smaller than the peak learning rate of the original run. We also turning off the weight decay for Fine Tuning.

## C    THRESHOLD DIMENSION

For completeness, we summarize the result presented in (Larsen et al., 2022). Let $\mathbf{A} \in \mathbb{R}^{D \times d}$ be a random Gaussian matrix with columns normalized to 1, $\mathbf{w}_t \in \mathbb{R}^D$ be a weight configuration, and $S$ be the set in $\mathbb{R}^D$ (typically a sublevel set of the loss function). Then we are consider the quantity:

$$P_s \equiv \mathbb{P}\Big[ S \cap \big\{ \mathbf{w}_t + \mathrm{span}(\mathbf{A}) \big\} \neq \emptyset \Big]. \tag{C.1}$$

i.e. the probability that the Gaussian affine subsapce defined by $\mathbf{A}, \mathbf{w}_t$ intersects the set $S$. We are then interested in understanding the minimal dimension for which an intersection occurs with high probability or the threshold training dimension.

**Definition C.1** (Threshold training dimension). *The threshold training dimension $d^*(S, t, \delta)$ is the minimal value of $d$ such that $P_s \geq 1 - \delta$ for some small $\delta > 0$.*

Their main result bounds the threshold dimension of an affine subspace, such that this subspace intersects a target set $M$ with high probability. The proof uses Gordon's Escape Theorem (Gordon, 1988), and depends on the Gaussian width of the projected set $M$.

**Definition C.2** (Gaussian Width). *The Gaussian width of a subset $M \subset \mathbb{R}^D$ is given by:*

$$r(M) = \frac{1}{2} \mathbb{E} \sup_{\mathbf{x}, \mathbf{y} \in M} \langle \mathbf{g}, \mathbf{x} - \mathbf{y} \rangle, \quad \mathbf{g} \sim \mathcal{N}(\mathbf{0}, \mathbf{I}_{D \times D}).$$

The local angular dimension is then the Gaussian width of $M$ projected onto the unit sphere around the subspace offset:

**Definition C.3** (Local angular dimension). *The local angular dimension of a general set $M \subset \mathbb{R}^D$ about a point $\mathbf{w}$ is defined as*

$$d_{\mathrm{local}}(M, \mathbf{w}) \equiv r^2(\mathrm{proj}_{\mathbf{w}}(M)). \tag{C.2}$$

*where $\mathrm{proj}_{\mathbf{w}}(M)$ denotes projection of a set $M$ onto a unit sphere centered at $\mathbf{w}$:*

$$\mathrm{proj}_{\mathbf{w}}(M) \equiv \{ (\mathbf{x} - \mathbf{w}) / \|\mathbf{x} - \mathbf{w}\|_2 : \mathbf{x} \in M \}.$$

The threshold training dimension is finally obtained by taking $D$ minus the local angular dimension.

$$d^*(M, \mathbf{w}) = D - d_{\mathrm{local}}(M, \mathbf{w}). \tag{C.3}$$

Intuitively, this relation means that the closer one gets to the set, the larger it's projection on the surrounding unit sphere will be, and hence the lower the dimension of the subspace that will be required to hit the subspace with high probability.

For neural network loss landscapes, this local angular dimension is challenging to compute. However, an analytic bound for the threshold training dimension can be derived in the case of a quadratic loss function $\mathcal{L}(\mathbf{w}) = \frac{1}{2} \mathbf{w}^T \mathbf{H} \mathbf{w}$ where $\mathbf{w} \in \mathbb{R}^d$ and $\mathbf{H} \in \mathbb{R}^{D \times D}$ is a symmetric, positive definite Hessian matrix with eigenvalues $\{\lambda_1, \ldots, \lambda_D\}$. This loss function can be used as a second-order approximation of the loss landscape surrounding a minima. The lower bound on the local angular dimension of $S(\varepsilon)$ about $\mathbf{w}$ is given by:

$$d_{\mathrm{local}}(\varepsilon, R) = w^2\big( \mathrm{proj}_{\mathbf{w}}(S(\varepsilon)) \big) \gtrsim \sum_i \frac{r_i^2}{R^2 + r_i^2}, \tag{C.4}$$

where $r_i = \sqrt{2\varepsilon / \lambda_i}$.

## D    IMP-LRR SUBNETWORKS CAN BE RETRAINED FROM AN EARLY REWIND POINT

IMP with learning rate rewinding (IMP-LRR) has been shown to exceed the performance of standard fine tuning, and match the performance of IMP-WR (Renda et al., 2020).

Both IMP-LRR and IMP-WR can be used to find matching subnetworks. However, IMP-WR has a special property: it can be retrained from an iteration early in training.

Let $\tau$ denote a training step from which IMP-WR works (i.e., the onset of linear mode connectivity). Here we further show that a mask $\mathbf{m}$ that gives a matching subnetwork with learning rate rewinding will also give a matching subnetwork $\mathbf{m} \odot \mathbf{w}_\tau$, i.e., the subnetwork can be rewound back to $\tau$ and retrained to the same error.

The results presented in Fig. 8 show that IMP-LRR-obtained subnetworks are not only matching with learning rate rewinding, but can also be retrained from the same rewinding iteration as IMP-WR subnetworks, and obtain the same error. This is due to the stability up to perturbations at the rewind point. In other words, IMP-LRR subnetworks perturb the network at the rewind point within the stability limit (see Section 3.2 for more details).

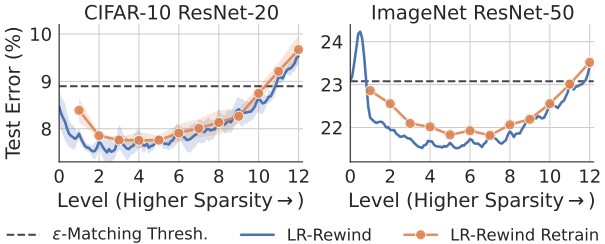

Figure 8: At each sparsity level (x-axis) we train the same sparse subnetwork obtained via IMP-LR under two conditions: we train the network with learning rate rewinding after pruning (standard IMP-LR, blue line), or we train the network by first rewinding it to a stable rewind point (like in standard IMP-WR, orange line). We evaluate the error of the resulting trained network (y-axis). The overlap between the two curves suggests that IMP-LR mask can be retrained from a stable rewind point and match the performance of standard retraining with learning rate rewinding. In other words, IMP-LR produces masks that can be retrained from a point early in training.

## E    LEARNING RATE REWINDING VS. FINE TUNING

We refer to FINE TUNING as training the pruned final model with the same final learning rate. Renda et al. (2020) show that fine tuning underperforms IMP-WR in final test accuracy. To alleviate this, Renda et al. (2020) propose a middle ground between IMP-WR and FINE TUNING called IMP with LEARNING RATE REWINDING (IMP-LRR). In IMP-LRR, the final pruned model is trained for the same amount of time as the original training run and with the original learning rate schedule, but starting at the weights at convergence instead of rewinding to $\mathbf{w}_\tau$. As shown in (Renda et al., 2020), this produces networks that perform equivalently to IMP-WR.

Fig. 9 compares fine tuning and IMP-LRR on CIFAR-10 and ImageNet. We show that IMP-LRR achieves a lower test error than fine tuning across sparsities, which can be explained by weight reequilibriation as IMP-LRR maintains a larger fraction of small weights compared to fine tuning.

## F    A SIMPLE ADAPTIVE PRUNING HEURISTIC FOR OPTIMIZING IMP

One of the limitations of IMP is its computational intensiveness. Pruning standard networks on standard benchmarks to the maximum sparsity achievable by IMP often involves retraining the network more than 10 times. For example, with standard hyperparameters and a 20% pruning ratio at each level, a ResNet-18 trained on CIFAR-100 requires 11 levels of pruning to achieve a sparsity of 9% weights remaining. Note, we specifically select this example because, among our experimental settings, a ResNet-18 trianed on CIFAR-100 was able to train to the lowest fraction of weights remaining and required the most number of levels; a speedup in this setting would be the most effective.

Part of the cost of IMP is due to the fact that 20% is not the optimal pruning ratio at every level—at low sparsity levels the network can be pruned more aggressively and at high sparsity levels, the pruning ratio must be small to not overshoot. 20% is often chosen as a balance between the high

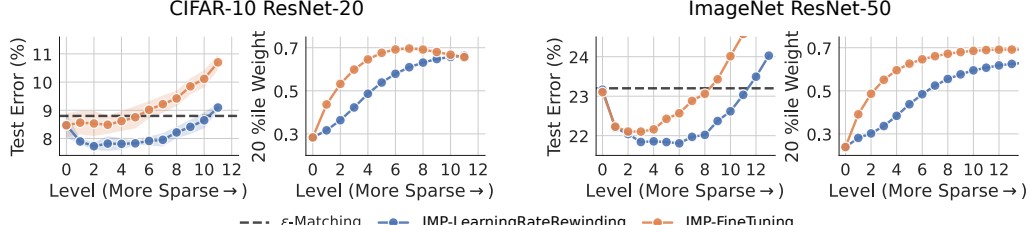

Figure 9: Learning rate rewinding yields trainable matching networks and provides small weights to prune further, unlike finetuning. The left plots in each dataset panel show the test error achieved by learning rate rewinding (LRR) and fine tuning at each level. The right plots in each dataset panel show the magnitude of the 20th percentile weight (normalized by the mean weight at that level so that we can compare across levels and strategies) at each level for both LRR and finetuning. With LRR, the 20th percentile weight is much smaller compared to finetuning, indicating LRR creates new small weights to further prune in a way that fine tuning cannot. In the retrain step, if we finetune, the distribution of weights doesn't change much from the pruned model and there are no small weights to prune (see also Fig. 7 and 22). This results in a large projection distance and thus we cannot find an intersection between the axial subspace and the LCS-set. On the other hand, with learning rate rewinding, the network weights achieve a new equilibrium and there will be additional new small weights to prune (see also Fig. 7 and 22). This explains why LRR is able to achieve higher accuracies.

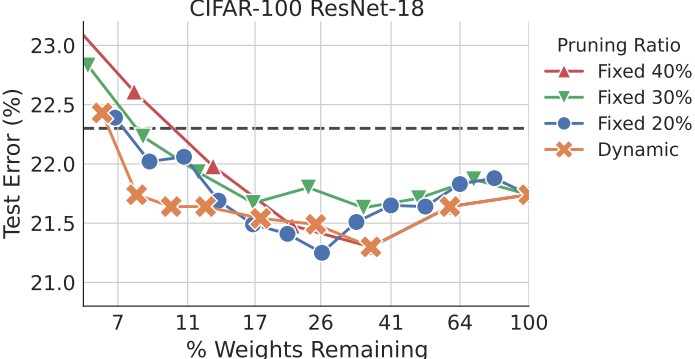

Figure 10: Adaptive Pruning Heuristic: A simple heuristic (described in Appendix F) for adapting the pruning ratio at each level (orange crosses) can yield a 33% improvement in training compute compared to IMP with a fixed pruning ratio of 20% at each level (blue circles).

pruning ratios possible at low sparsity levels and the low pruning ratios necessary at high sparsity levels since it is not evident a priori what the appropriate pruning ratio at any given level is.

Insights from our investigation of the error landscape at each IMP step (Fig. 3) suggests a natural heuristic for determining the optimal pruning ratio; choose a pruning ratio that keeps the pruned solution within the LCS-set of the unpruned solution. This guarantees that we will find an axial subspace that intersects with the desired LCS-set and at low sparsity levels, this effectively allows us to prune larger fractions of the weights. However at higher sparsities, this heuristic may result in multiple iterations of pruning a small fraction of the weights as even small perturbations take us out of the LCS-set. Additionally, we don't have access to the test error at training time and so cannot directly determine when we have pruned out of the LCS-set. To overcome these challenges, we design our heuristic as follows:

1. Estimate an appropriate $\varepsilon$ for a linearly connected *training loss* sublevel set. We estimate this using the standard deviation of the training loss across batches over the last epoch of training the dense network. We will refer to this as the train LCS-set. Our threshold for leaving the train LCS-set is thus the training loss of the dense network + the estimated $\varepsilon$.

2. At the Level $L$ solution, we sweep pruning ratios of 10%, 20%, ... to find the maximum pruning ratio $p$ such that level $L$ solution after pruning remains within the train LCS-set, i.e. the training loss along the linear path between the Level L solution before and after pruning is less than the threshold estimated in step 1.

3. We will use the $p$ found in step 2 as the new pruning ratio of IMP in this level $L$ (if $p > 0.2$, or otherwise we will use 20% as the pruning ratio), i.e. we prune $\max(p, 0.2)$ of the smallest magnitude weights and start the retraining step of IMP.

As demonstrated in Fig. 10, this heuristic of choosing a dynamic pruning ratio indeed reduces the compute by approximately 33%—we only need to retrain 7 instead of 11 times without losing any accuracy, and at each level, the cost of determining the pruning ratio involves just a few forward passes through (possibly a random subsample of) the training set.

This heuristic is especially effective when a pruning ratio of 20% is much smaller than the optimal pruning ratio. However, in datasets suchs as ImageNet, the improvement is moderate because the optimal pruning ratio at early levels is already close to 20%.

Note: we do not claim that this is an optimal algorithm, our goal in this work is to achieve a deeper scientific understanding of IMP, this heuristic is just one of the many ways we can use these insights to improve the algorithm. We leave the task of fully exploring and characterizing these avenues to future work.

## G   PRUNING A DENSE NETWORK

In our experiments investigating linear mode connectivity of trained IMP solutions at different levels (Fig. 2 and 16), we find that the Level 0 (dense) solution is separated from the solutions at higher levels by a small but non-zero error barrier. In fact, for rewind steps at which we can find matching sparse networks of high sparsity, there always exists a piecewise linear path that interpolates between solutions at successive levels with 0 error barrier. Only for CIFAR-10, does this extend to the level 0 solution.

An interesting question that challenges our hypotheses is, why are we not connected to the dense network? The answer lies in Fig. 4—the dense network is not robust to perturbations at the rewinding step used in our experiments. When we perturb the dense network at the rewind point by a distance equal to the norm of the projection induced by the level 1 mask and train in the full dense space, the resulting solution is not linearly mode connected to our original level 0 solution. In fact, at this rewind step, the dense network isn't even linearly mode connected (robust to SGD noise); Frankle et al. (2020a) find that IMP can find matching solutions after the onset of LMC in the sparse axial subspace which typically occurs before the onset of LMC in the dense space. As observed in Figure 16, as the rewind step increases, the error barrier between level 0 and level 1 gets smaller.

This result further underscores the importance of robustness to perturbation at the rewind step that we observe in Fig. 4. At level 0, even though we have an 80% sparsity axial-subspace that intersects the LCS-set of the level 0 solution, since the network is not robust to perturbation at the rewind step, we train to a level 1 solution in a different basin with a small error barrier to the level 0 solution.

However, this begs the question, why at level 1 do we not need to train into the same LCS-set as the level 0 solution? At level 1, we are still in a massively over-parameterized regime with 80% of the weights remaining. We can thus find a solution in this space that is as good as the solution at level 0. Indeed if the weights are rewound to initialization, which is essentially equivalent to having a random network with 80% sparsity, we get the same error Fig. 15. This is not true at high sparsities—if at a later level, we are unable to find an intersection with the LCS-set of a matching network, then it is difficult to find a different matching solution in that sparse space as sparse training without explicit knowledge of the end-point is still a difficult task.

What is surprising is that, the actual error barriers between a level 0 and level 1 IMP solution (blue point in Fig. 4 at level 0) is actually much smaller than the error barrier between the original level 0 network and a randomly perturbed network (orange point in Fig. 4 at level 0). Thus the IMP mask projection creates a perturbation that is more stable than a random perturbation. Investigating why this is true is left to future work.

## H  ERROR LANDSCAPE GEOMETRY OF ITERATIVE GRADIENT PRUNING

In this section, we show that our key observation, all matching networks are in the same LCS Set, holds for Iterative Gradient Pruning (IGP). IGP is a common iterative pruning strategy similar to IMP, except, at each pruning step, instead of pruning weights with the smallest magnitude, we prune weights with the lowest absolute value of (weight × gradient) (Blalock et al., 2020). Fig 11 shows the results of using IGP instead of IMP in the same setting as in Fig. 2. IGP does not perform as well as IMP—it ceases to produce matching subnetworks at earlier levels of pruning. However, pairs of solutions at successive levels do not have an error barrier between them and all matching solutions are in the same LCS set when IGP is rewound to step 2000. This suggests that finding pruned solutions in the LCS set of the dense solution might be a general feature of iterative pruning algorithms that start with a trained dense network and consrtuct nested subnetworks at each iteration.

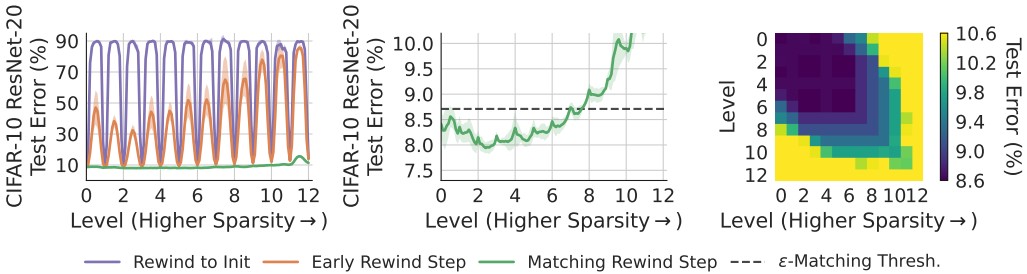

Figure 11: Same plots as Fig. 2 but for Iterative Gradient Pruning (IGP) on CIFAR-10 ResNet-20. 20% of the weights are pruned at each step. Though IGP stops being matching at lower levels compared to IMP, the matching networks all lie in the same LCS set.

To further investigate the reason why the performance of IGP is inferior to IMP, we calculated the Hessian along the directions given by three different pruning algorithm—Magnitude Pruning, Random Pruning and Gradient Pruning. (Fig. 12) We found that both Magnitude Pruning and Gradient Pruning found flatter directions than Random Pruning. In fact, Gradient Pruning found directions of similar sharpness (Hessian along the given direction) to Magnitude Pruning. However, because Magnitude Pruning aims to prune the smallest magnitude weights, the fractional distance (defined in Eq. I.1) of the projection is smaller than the one given by Gradient Pruning. According to our theory (Sec. 3.3), IGP would break down before IMP.

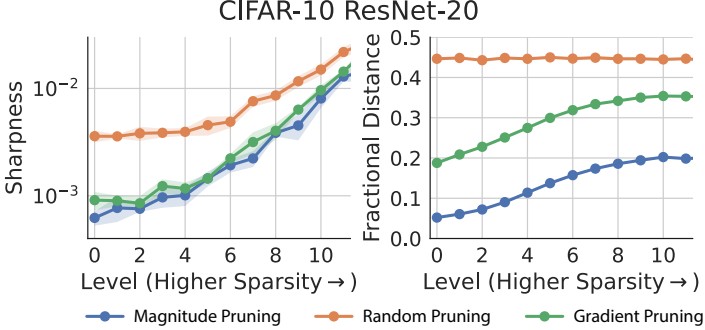

Figure 12: Gradient Pruning direction is of similar sharpness to Magnitude Pruning direction but its fractional distance is larger than Magnitude Pruning. We looked at three different pruning algorithms at each level of an IMP experiment. We compute the Hessian along the pruning direction on the training dataset (left) and the fractional distance as defined in Eq. I.1(right). Although Gradient Pruning can find similar flat direction, its fractional distance is larger, resulting in the poor performance of IGP compared with IMP, which is expected from our theory in Sec. 3.3. All results were averaged over four replicates.

# I ADDITIONAL EXPERIMENTS ON THE ROBUSTNESS OF SGD

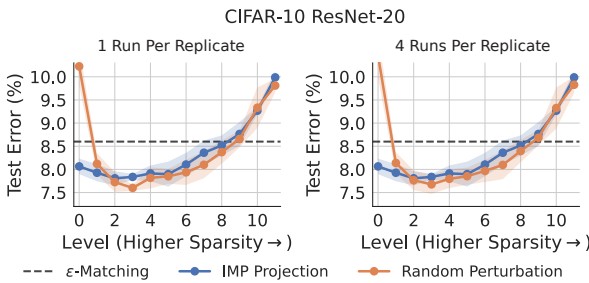

Figure 13: **Sensitivity of SGD Robustness to Number of Runs.** For CIFAR-10 ResNet-20 we repeat the experiment in Fig. 4 but with more runs to test its sensitivity. All experiments are conducted for 4 replicates that use independent seeds. Left: identical to the middle plot in Fig. 4, reproduced for comparison. One random perturbation was performed per replicate, means and standard deviations are calculated across the 4 replicates. Right: We use 4 independent random perturbations per replicate. The mean and standard deviation of the orange line is calculated across a total of 16 runs (4 runs each for 4 replicates). The results are extremely similar suggesting that exploring one random perturbation per replicate is reasonable given computational constraints.

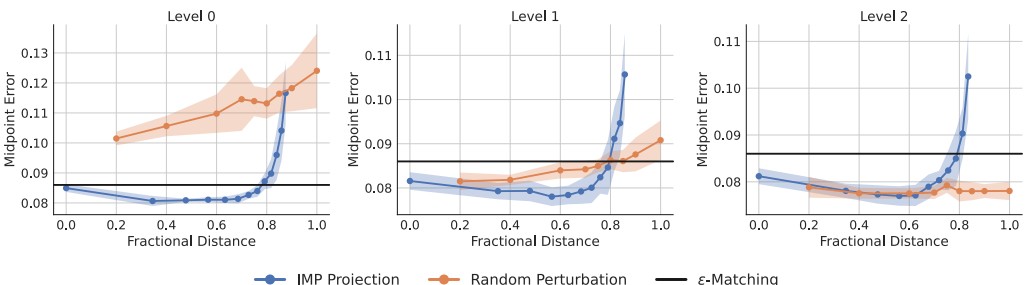

Figure 14: **SGD Robustness to Perturbations of Varying Lengths.** On CIFAR-10 ResNet-20, for the first 3 levels of IMP (in the titles), we look at the robustness of SGD to random perturbations of various lengths compared to IMP mask projections to subsequent levels. Fractional distance on the x-axis is measured as the ratio of norms of the perturbation/projection and the weights at the rewind point. Midpoint error is measured as the error at the midpoint on the line in weight space connecting the trained network from the rewind point and the trained perturbed/projected network. Surprisingly, at high sparsities, SGD is robust to large random perturbations.

In this section, we further explore the robustness properties of SGD. Recall that in Section 3.2, we found that, when the rewind point is projected onto the new axial subspace, SGD finds the desired LCS set of the previous IMP solution with high probability because even a random perturbation of the same magnitude will train to the LCS set with high probability. Thus, the ability of SGD to extract the location of the LCS set from the mask follows from the general robustness of SGD late in training.

In Fig. 13 we verify that our results are still valid if we perform more random perturbations and in Fig. 14 we look at the effects of the random perturbation distance at a given sparsity level on the robustness of SGD.

**Experiment 1, Fig. 13:** In this experiment on CIFAR-10 ResNet-20, we add more random perturbations to our experiments to verify that our results hold in general and are not an artifact of insufficient sampling. Fig. 13 left is identical to Fig. 4 middle column, we reproduce it here for comparison. Fig. 13 right is the new experiment with more perturbations. Both figures use 4 independently trained replicates to calculate the mean and standard deviation of both the IMP projection

and the random curves. In the left plot, for each replicate we only sample one random projection. On the right plot, for each replicate we sample 4 independent random projections. For the orange curve, we then average over all the random projections for each replicate for a total of 16 runs. The blue curves are still the 4 replicates for IMP; there is no additional degree of freedom to sample over. The experimental setup is otherwise the same as described in Fig. 4 and section 3.2. The results are very similar suggesting that the robustness of SGD is indeed a true phenomenon and not an artifact of insufficient sampling.

**Experiment 2, Fig. 14:** We now look at the dependence of the robustness of SGD to random and IMP perturbations on different lengths at early IMP levels for CIFAR-10 and ResNet-20. In Fig. 14, the 3 columns correspond to experiments performed at the first 3 sparsity levels: level 0 = 100% weights remaining, level 1 = 80% weights remaining, and level 2 = 64% weights remaining. Since IMP creates a sequence of nested subspaces, we can examine at any level $L$ the projection of the rewind point onto the axial subspaces of levels $L + 1$, $L + 2$, and so on. The blue curves in the figure correspond to these IMP projections. Specifically, at level $L$, denoted in the title, consider the projection distance of $\mathbf{m}^{(L)} \odot \mathbf{w}_\tau^{(L)}$ onto the axial subspaces of $\mathbf{m}^{(L+1)}, \mathbf{m}^{(L+2)}, ....$ The x-axis corresponds to these distances as a fraction of the norms of the weights given by

$$\frac{\|\mathbf{m}^{(L)} \odot \mathbf{w}_\tau - \mathbf{m}^{(L+k)} \odot \mathbf{w}_\tau\|_2}{\|\mathbf{m}^{(L)} \odot \mathbf{w}_\tau\|_2} \tag{I.1}$$

for $k = 1, 2, 3, ....$ The y-axis is the test error at the midpoint interpolation between the corresponding IMP solutions $\mathbf{w}^{(L)}$ and $\mathbf{w}^{(L+k)}$ for $k = 1, 2, 3, ....$ Not this is different from Fig. 4 as instead of looking at the midpoint between solutions at subsequent levels, we are looking at the midpoints between the solution at the current level and all following levels. The black lines correspond to the matching threshold: if a point is below that line, it means that the two corresponding end points are in the same LCS-set. The blue curves show how the mask projection distance at the rewind point affects whether two networks at different levels train into the same LCS-set. As expected, they do until we reach the last few levels. The mean and standard deviation is calculated across 4 independent replicates.

The orange curves show the relationship between the length of random perturbations at the rewind point in the $\mathbf{m}^{(L)}$ space and whether the networks before and after the perturbations train to the same LCS-set. Specifically, at each level $L$, denoted by the title, we apply a random perturbation $\mathbf{v}$ of length $c$ to $\mathbf{m}^{(L)} \odot \mathbf{w}_\tau$. The length $c$ is chosen to have the correct fractional length in the x-axis and the direction is chosen uniformly from all directions in the $L$-dimensional axial subspace given by $\mathbf{m}^{(L)}$. This perturbed network is then trained in the $\mathbf{m}^{(L)}$ subspace for $T - \tau$ steps to get $\tilde{w} = \mathcal{A}_\tau(\mathbf{m}^{(L)} \odot \mathbf{w}_\tau + \mathbf{v}, T - \tau)$. The y-axis plots the test error at the midpoint between this trained network $\tilde{w}$ and the IMP solution at $\mathbf{w}^{(L)}$. For points along the orange curve that are below the black line, the network obtained from applying a random perturbation of the given length to the level $L$ rewind point will train into the LCS-set of the level $L$ IMP solution. The mean and standard deviation is calculated across 4 independent replicates.

For the higher levels, we find that the LCS set the network trains into is robust to large perturbations at the rewind point. In fact, the network is robust to random perturbations that are larger than the IMP mask projections of all successive matching levels. This provides further evidence for the role of SGD robustness in enabling the pruned network to train back into the LCS set of the previous IMP solution. Note that an IMP mask projection of a given length may not find a matching solution because the subspace corresponding to the mask may not intersect with the desired LCS-set. However, a random perturbation of the same length might train into the desired LCS-set as training is happening in the higher-dimensional space that we know contains this set. Also the results from Level 0 have large error barrier between the networks before and after random perturbation for the same reason as described in Appendix G.

## J    FULL RESULTS

Here we present the full set of experiments performed for the results in the main text as well as extensions to additional datasets.

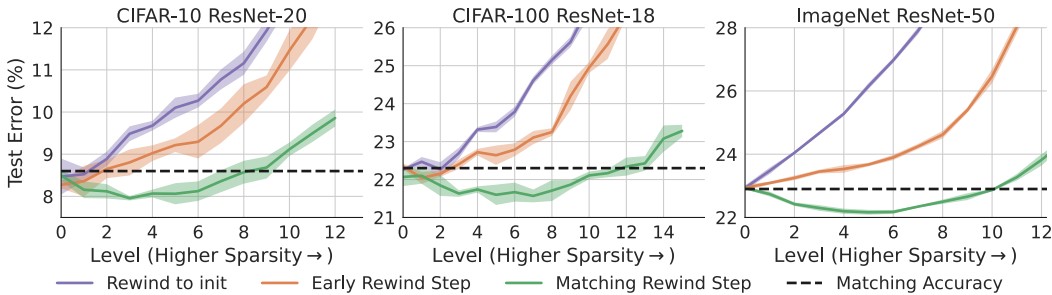

Figure 15: Error and Rewind Steps for IMP. Purple curves show rewind step 0. Orange curves show early rewind steps (250 for CIFAR-10, 400 for CIFAR-100, 1250 for ImageNet) and the green curves show rewind steps that produce sparse matching subnetworks (step 2000 for CIFAR-10, 3200 for CIFAR-100 and 5000 for ImageNet).

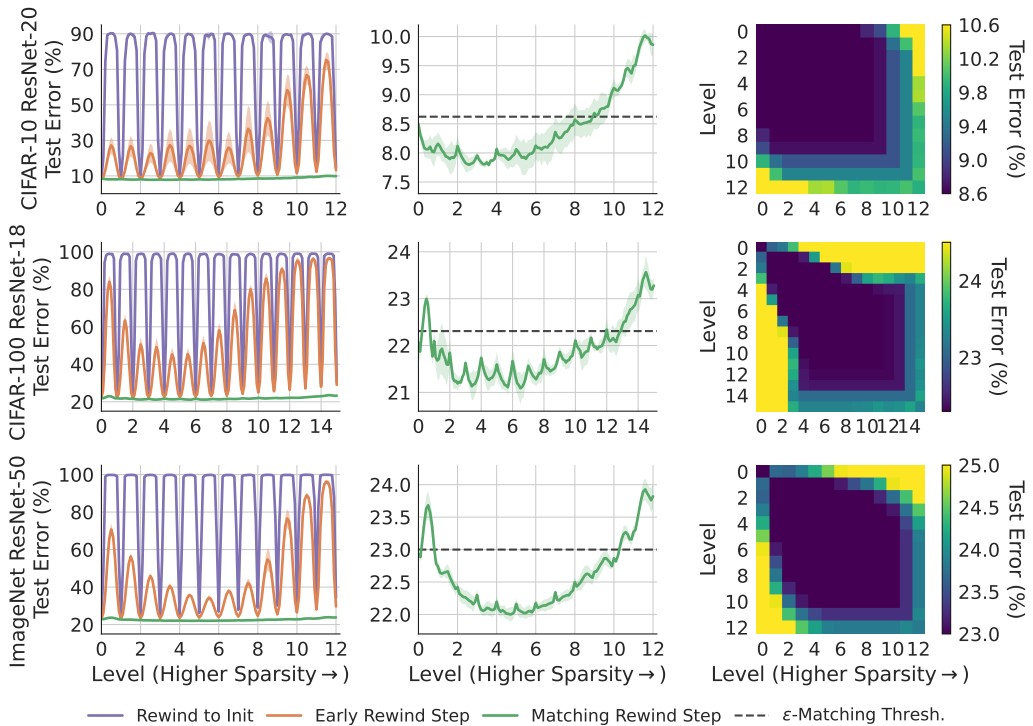

Figure 16: The same plots as Fig. 2 on CIFAR-10 ResNet-20, CIFAR-100 ResNet-18, and ImageNet ResNet-50.

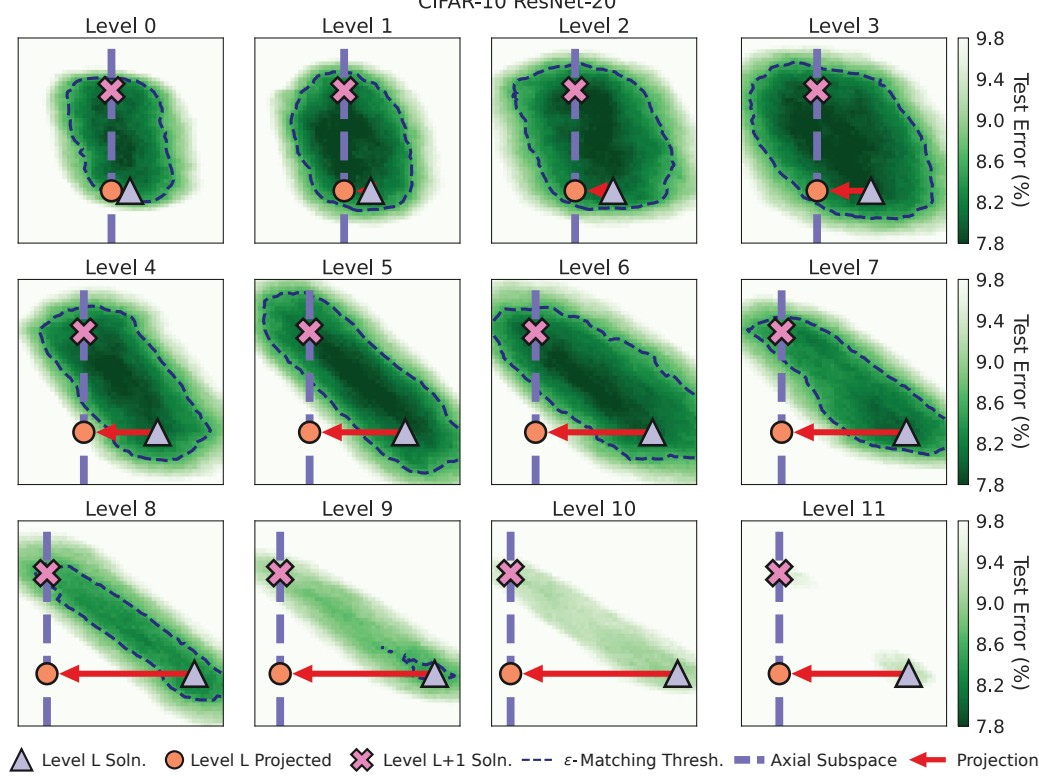

Figure 17: All the projection levels for CIFAR-10. The two-dimensional slice of the loss landscape determined by at each level $L$ by 3 points: the solution found at level $L$ (grey triangle), the projection of this solution (orange circle) onto the axial subspace determined by the smallest weights (blue dotted line), and the solution found at level $L + 1$ by retraining with the mask (pink cross). Note that by definition, the $L + 1$ solution must also lie on the axial subspace. The light dotted black line outlines the $\varepsilon$ sub-level set.

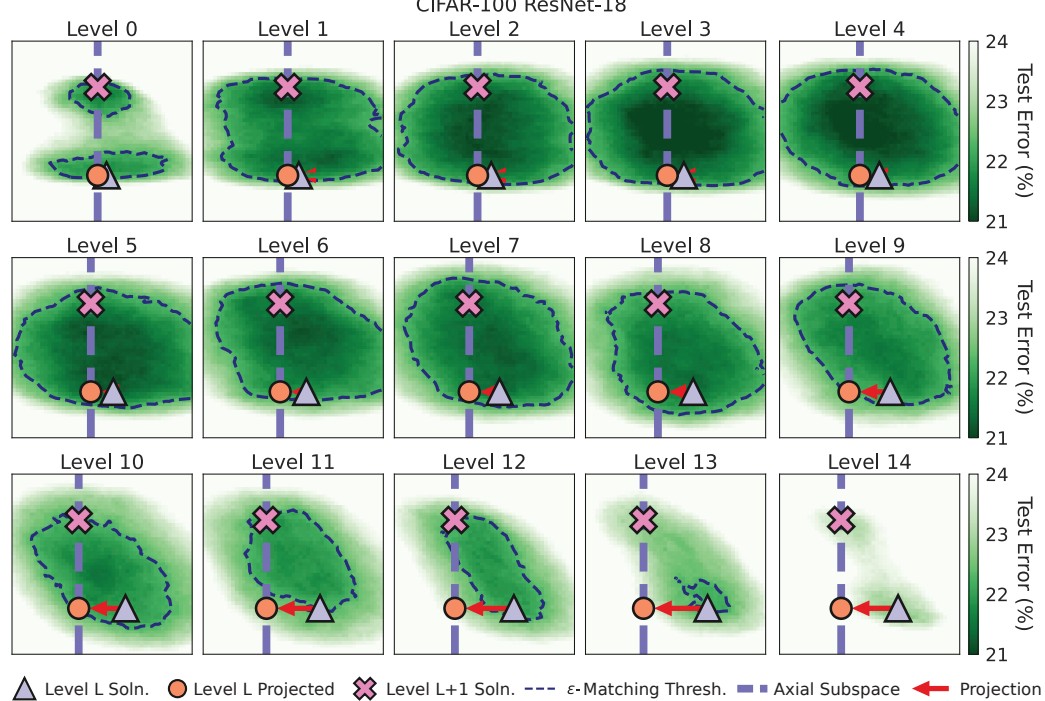

Figure 18: All the projection levels for CIFAR-100. The two-dimensional slice of the loss landscape determined by at each level $L$ by 3 points: the solution found at level $L$ (grey triangle), the projection of this solution (orange circle) onto the axial subspace determined by the smallest weights (blue dotted line), and the solution found at level $L + 1$ by retraining with the mask (pink cross). Note that by definition, the $L + 1$ solution must also lie on the axial subspace. The light dotted black line outlines the $\varepsilon$ sub-level set.

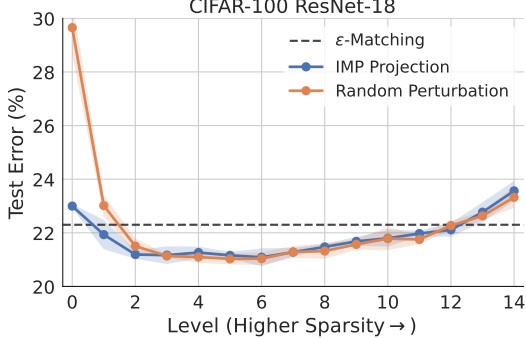

Figure 19: Stability to perturbations at the rewind point for CIFAR-100.

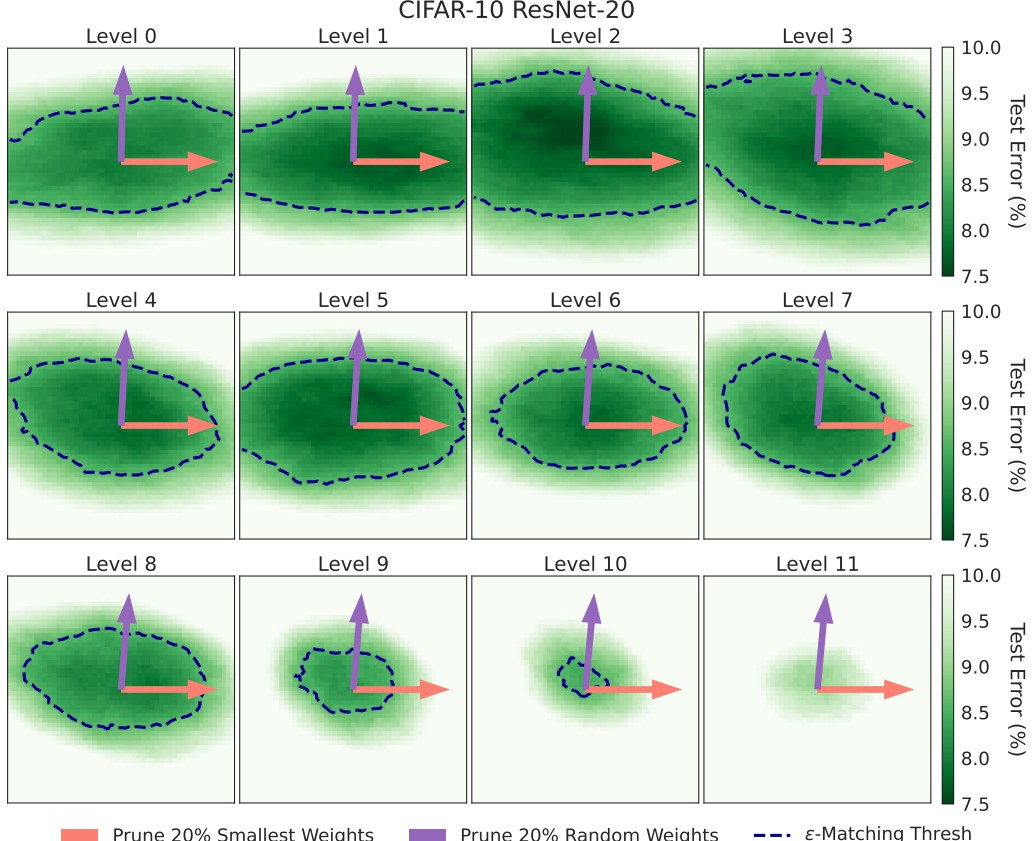

Figure 20: The same plots as Fig. 6 but for CIFAR-10 dataset. Projection of the error landscape in the subspace spanned by a random projection and small magnitude weight projection. Flat error directions correlate with small magnitude directions. The size of the LCS-set decreases with increasing sparsity levels.

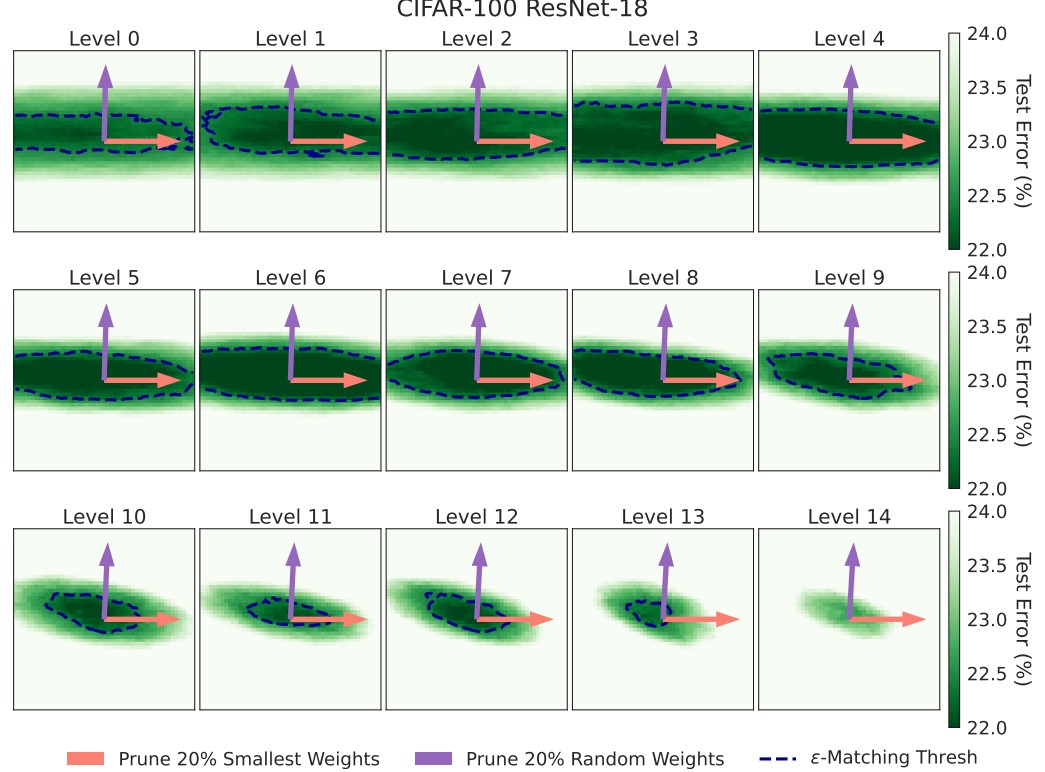

Figure 21: The same plots as Fig. 6 but for CIFAR-100 dataset. Projection of the error landscape in the subspace spanned by a random projection and small magnitude weight projection. Flat error directions correlate with small magnitude directions. The size of the LCS-set decreases with increasing sparsity levels.

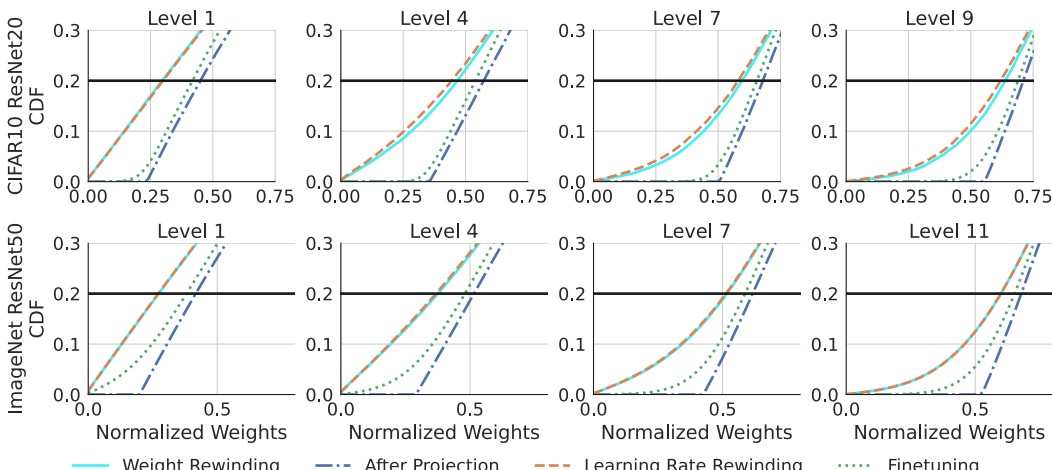

Figure 22: The same plots as Fig. 7 on CIFAR-10 ResNet-20 and ImageNet ResNet-50. The zoomed out version is shown in Fig. 23

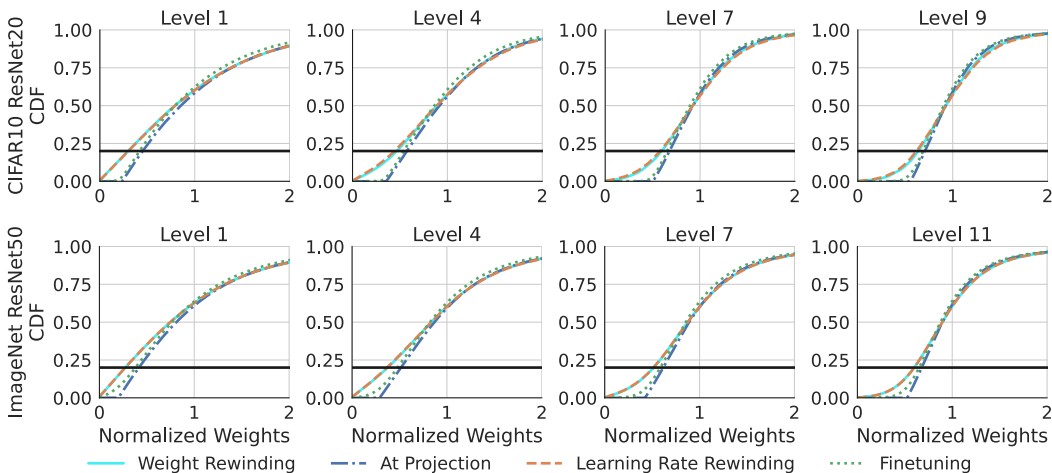

Figure 23: The same plots as Fig. 22 but showing the zoomed out version.

