# OpenReview forum: "Unmasking the Lottery Ticket Hypothesis: What's Encoded in a Winning Ticket's Mask?"
_ICLR.cc/2023/Conference — ICLR 2023 notable top 25%_

### Official Review · Reviewer_d22d · 2022-10-21

**Confidence:** 3
**Correctness:** 4
**Technical Novelty And Significance:** 3
**Empirical Novelty And Significance:** 3
**Recommendation:** 8

**Clarity, Quality, Novelty And Reproducibility:**

The paper is well written and the experiments presented to validate the authors' claims are well described. As said in the previous section, I think that it can be a significant contribution to the research in this field.

**Strength And Weaknesses:**

I think that this paper is very interesting because it tries to answer to some fundamental questions related to the lottery ticket hypothesis, providing a significant contribution to the research in this field. The analysis presented in this paper is based on solid theoretical assumptions and validated by extensive empirical experiments.

**Summary Of The Paper:**

This paper shed some light on the underlying principles that allow IMP to find winning tickets. In particular, the authors focus on four fundamental questions regarding the information provided by the mask, the need of an iterative procedure for pruning, and the differences between retraining, learning rate rewinding and finetuning.

**Summary Of The Review:**

I think that this is a solid paper and it provides a significant contribution to explain the underlying principles of the lottery ticket hypothesis.

---

> ### Author Response · Authors · 2022-11-15
> **Response to Reviewer d22d**
>
> Thank you for your enthusiasm and encouraging comments; we hope that you will continue to champion our work.  In response to the feedback of all reviewers, we have made the following changes to improve the paper:
> * We have added a summary of the primary related work to the main body of the text.
> * We have added comparisons to Iterative Gradient Pruning in Appendix H to show the results hold for other iterative pruning methods.
> * We have added additional experiments on the robustness of SGD in Appendix I (the primary results are in Section 3.2).

---

### Official Review · Reviewer_vB9o · 2022-10-22

**Confidence:** 2
**Correctness:** 4
**Technical Novelty And Significance:** 3
**Empirical Novelty And Significance:** 3
**Recommendation:** 8

**Clarity, Quality, Novelty And Reproducibility:**

**Quality**

Aside from my concerns in the "Strengths and Weaknesses" about the amount of experiments relative to the number of claims made in the paper, I believe the experiments are of high quality.

**Clarity**

The work is clear and is well written.

**Originality**

As far as I can tell the work is original and places itself well among the literature.

**Comments/Additional feedback:**

On page 4 when referencing Figure 2 the “red” curve as you describe it appears to be more orange colored in the figure.  Perhaps you could recolor the figure or change the description.

Second paragraph of the discussion in Section 4 references multiple results in the Appendix, e.g.
“we show in Fig. 10 that it is possible to achieve the same performance as IMP but with fewer levels of pruning by dynamically choosing the per-level pruning ratio so as to stay within the LCS-set after projection”.  It may be more clear to the reader to emphasize that these results appear in the Appendix but not the main body.

**Strength And Weaknesses:**

**Strengths:**

The paper investigates the relevant problem of identifying the conditions for the success of IMP.  The hypothesis is appealing and is consistent with their data.  The figures are clear and readable.

**Weaknesses:**

This paper lacks an adequate discussion of related work in the main body.  While they have a related work section in the Appendix, it is standard to have this section in the main body.  It is important for readers of the work to have these comparisons made clear.  Also note that reviewers for ICLR are not required to read the supplementary material for better or for worse, which makes it even more important for this section be included in the main body.

The paper makes many key claims (e.g. the six bullet points spanning the end of page 3 and the start of page 4).  While there are data to support each of these claims, these are big proclamations and some could be a separate paper on their own.  For example when investigating robustness of SGD you would need to take many perturbations and then train to really see if all the solutions are linearly connected due to the high dimensionality of the space.  I think the paper could potentially be improved if it slightly reduced its scope and focused the experiments on its most important claims.


**Summary Of The Paper:**

This paper investigates why iterative magnitude pruning (IMP) is successful.  The authors hypothesize that the masks discovered during iterations of IMP encode subspaces that intersect the linearly connected loss sublevel sets associated with the previous iterate.  They present multiple experiments to support this hypothesis.

**Summary Of The Review:**

I would be inclined to accept this paper after revisions are made to properly address the literature in the main body of the paper and to make proper comparisons to the present work.

---

> ### Author Response · Authors · 2022-11-15
> **Response to Reviewer vB9o**
>
> We thank the reviewer for their comments on our paper. In response, we have added a summary of primary related work to the main text and expanded our experiments related to the robustness of SGD.  We address the reviewer’s specific concerns below:
>
> > This paper lacks an adequate discussion of related work in the main body. While they have a related work section in the Appendix, it is standard to have this section in the main body.
>
> We have added a section on the primary related works at the end of the introduction.
>
> > The paper makes many key claims (e.g. the six bullet points spanning the end of page 3 and the start of page 4). While there are data to support each of these claims, these are big proclamations and some could be a separate paper on their own.
>
> We have included the full set of results in a single paper because we think that each piece is important to telling a cohesive story about IMP through the error landscape geometry.  However, we agree with the reviewer that additional clarity could be provided for the point related to the robustness of SGD.  We outline the changes made in the next response.
>
> > For example when investigating robustness of SGD you would need to take many perturbations and then train to really see if all the solutions are linearly connected due to the high dimensionality of the space. I think the paper could potentially be improved if it slightly reduced its scope and focused the experiments on its most important claims.
>
> We have addressed this concern in two ways.  First, we have reworded the contribution to more accurately describe our results.  We had previously said “any two networks separated by a distance equal to that of successive rewind points will train to the same linearly connected mode.”  This should have been worded as random perturbations of this distance are *likely* to produce networks with this property.  We now say “The linearly connected modes these networks train into are not only robust to SGD noise, but also to random perturbations of length equal to the distance between rewind points at successive levels”, and “two pruned rewind points at successive sparsity levels are *likely* to navigate back to the same linearly connected mode yielding matching solutions.”
>
> Second, we have added additional experiments exploring this phenomenon further in Appendix I.  Figure 12 shows that doing more runs per replicate has minimal effects on the results for CIFAR-10, and Figure 13 shows that the results of sweeping over the perturbation distance for the first few levels of sparsity for CIFAR-10.
>
> > On page 4 when referencing Figure 2 the “red” curve as you describe it appears to be more orange colored in the figure. Perhaps you could recolor the figure or change the description.
>
> Thank you for catching this.  We had different descriptions of the curve in the figure caption and in the main text; we have changed the descriptions to match such that both now call the curve “orange.”
>
> > Second paragraph of the discussion in Section 4 references multiple results in the Appendix, e.g. “we show in Fig. 10 that it is possible to achieve the same performance as IMP but with fewer levels of pruning by dynamically choosing the per-level pruning ratio so as to stay within the LCS-set after projection”. It may be more clear to the reader to emphasize that these results appear in the Appendix but not the main body.
>
> We have added a reference to Appendix F here to clarify that this figure and result are described in the appendix.
>
> > I would be inclined to accept this paper after revisions are made to properly address the literature in the main body of the paper and to make proper comparisons to the present work.
>
> As stated above, we have edited the paper to include a related works section in the main body of the text at the end of the introduction.
>
> We would appreciate it if the reviewer would let us know if the changes we have made addressed all their concerns adequately and if they now consider the paper to make a proper comparison to previous work.

---

> > ### Comment · Reviewer_vB9o · 2022-11-15
> > **Initial response to authors**
> >
> > I have reviewed the changes.  Now the authors have a related work section in the main body which addresses the literature, as well as a link to further discussion in the Appendix.  They have also rephrased their claims about the robustness of SGD to be probabilistic and have added new experimental data to demonstrate the robustness of SGD across 16 runs.  Given the high economic cost of these experiments I think 16 independent runs is a reasonable number.  The authors have addressed my concerns and **I am raising my recommendation to an 8 accept rating to reflect the improved quality of the manuscript.**

---

### Official Review · Reviewer_y14k · 2022-10-23

**Confidence:** 4
**Correctness:** 3
**Technical Novelty And Significance:** 3
**Empirical Novelty And Significance:** 4
**Recommendation:** 10

**Clarity, Quality, Novelty And Reproducibility:**

This is a brilliant paper, clear and novel, reviewing it is an honor and a pleasure. The results should be easily reproducible.

I do, however, have a few questions to the authors, as well as some suggested corrections. Please see below.

“flatter error landscapes allow more aggressive pruning” — isn’t this an obvious consequence of flatter landscapes having more irrelevant parameters (hence the flatness, i.e. no or little influence on the loss)?

Page 6: “But how is SGD able to extract this information from the mask at the rewind point.” - Should end with a question mark.

The authors speak of the “robustness of SGD training to perturbations at late enough rewind steps”. Would it be valid to say that rewinding late enough does not actually cause the algorithm to leave the attraction basin? I think the attraction basin perspective is necessary here, otherwise the said robustness seems arbitrary. From the loss landscape perspective, what causes the robustness?

“This happens because the error landscape is getting sharper at higher sparsity levels” - why is the landscape getting sharper at higher sparsity levels? And can it thus be concluded that self-regularized solutions (“lottery tickets”) would typically be found in sharp minima? The latter would be a very interesting observation, since current theory favors flat rather than sharp minima (at least in terms of the expected generalization performance).

**Strength And Weaknesses:**

Strengths:
* The paper studies IMP from the loss landscape perspective, the results are insightful and open up many future research directions.
* The obtained insights have a very practical application (e.g. improving IMP).
* The paper is very well written, the experiments are realistic.

Weaknesses:
* The paper does not link the observations to the attraction basins in the loss landscape, which to me feels like an important omission.
* The appendices for the paper are extensive, and include such sections as related work, which seem better suited in the main body of the paper. In general, I think this paper would be more suited for a journal rather than a conference publication. However, I acknowledge that the ML community strongly favors top-tier conferences, and understand authors’ decision to compress their work into 9 pages. I commend the impressive effort, but honestly believe that the narrative would have benefited from a less stringent page limit.

**Summary Of The Paper:**

The paper investigates iterative magnitude pruning (IMP) from the perspective of loss landscapes to answer questions such as why is it necessary to rewind the search iteratively rather than perform the pruning in one-shot fashion. The authors show that the IMP mask encodes the location of linearly connected modes (i.e. modes with low error barrier) containing matching sparse solutions. Further, the authors show that rewinding is necessary to “equalize” the weights to identify the next set of low-magnitude weights to prune. Finally, a link to the Hessian eigenvalue spectrum is established. The insights can lead to improved (faster) versions of IMP, one such version is proposed by the authors in the appendices.

**Summary Of The Review:**

To conclude, I would be very sad if this paper is rejected, since it is definitely one of the best ones that I have recently reviewed or read. I think this work is potentially very impactful, and I believe that the loss landscape perspective provides an excellent lense to deeper our understanding of neural networks training and behavior.

---

> ### Author Response · Authors · 2022-11-15
> **Response to Reviewer y14k**
>
> Thank you for your enthusiasm and encouraging comments; we hope that you will continue to champion our work.  We address your questions below.
>
> >The paper does not link the observations to the attraction basins in the loss landscape, which to me feels like an important omission.
>
> > The authors speak of the “robustness of SGD training to perturbations at late enough rewind steps”. Would it be valid to say that rewinding late enough does not actually cause the algorithm to leave the attraction basin? I think the attraction basin perspective is necessary here, otherwise the said robustness seems arbitrary. From the loss landscape perspective, what causes the robustness?
>
> Entering an attraction basin of the loss landscape at later epochs is indeed the key concept that underlies robustness of SGD. If we define an attraction basin of an LCS set as the set of all weights $w$ such that, if trained from $w$, the network will converge inside the LCS set, then “robustness of SGD to perturbations of size $c$” is equivalent to stating that for a given $w$, a random sample from a ball of radius $c$ lies in the same attraction basin as $w$ with high probability. We choose to use “robustness of SGD” because that is the quantity that we directly measure in our experiments in Figure 4 and 15. However, we agree that the concept of an attraction basin provides useful context and intuition so we mention this connection in the paper (see the new text following Definition 3.1).
>
> > “flatter error landscapes allow more aggressive pruning” — isn’t this an obvious consequence of flatter landscapes having more irrelevant parameters (hence the flatness, i.e. no or little influence on the loss)?
>
> It is indeed intuitive that flatter error landscapes would allow more aggressive pruning provided that the solution is close to an axial subspace. However, if the closest axial subspace is far from your solution, the local flatness may not be as informative: local flatness may describe how much the parameters can be perturbed in flat directions while maintaining the same error, but we may not be able to entirely zero-out those weights. Section 3.3 addresses this: under the assumption that $\varepsilon$-matching threshold is small enough that the local quadratic approximation of the loss landscape holds, we quantitatively estimate the probability that the LCS set intersects an axial subspace as a function of the local Hessian eigenspectrum and the distance traveled due to pruning. Our contribution adds a mechanistic understanding to the intuition that flatter landscapes have more irrelevant parameters and so we should be able to prune them more aggressively. We have reworded the contributions section to more accurately reflect this.
>
>
> > This happens because the error landscape is getting sharper at higher sparsity levels” - why is the landscape getting sharper at higher sparsity levels? And can it thus be concluded that self-regularized solutions (“lottery tickets”) would typically be found in sharp minima? The latter would be a very interesting observation, since current theory favors flat rather than sharp minima (at least in terms of the expected generalization performance).
>
> This is indeed an interesting observation and something we are excited to pursue further. It is perhaps expected that the landscape gets sharper at higher sparsity levels given the following intuition. Note that pruning more parameters increases the projection distance. Therefore, by our observations in Section 3.3, we want to make sure that when we prune, we move in flat directions to maximize the probability of our axial subspace intersecting the LCS set. However, this comes at a cost: at every matching pruning level, we stay in the same basin while pruning a percentage of the flattest directions; if the Hessian captures the flat directions well, after pruning the remaining non-pruned directions are the sharper ones. So at every level, the loss function at the intersection of the LCS set and our axial subspace will get sharper.
>
> The sparsest lottery ticket does indeed end up in a sharp basin. However, the key thing to note is that flatness in the full space is the error landscape property that allows us to heavily decrease the parameter count: had we started in a sharp basin, we would not have been able to prune to the same level of sparsity. So in this sense, our findings are consistent with current consensus on flat vs sharp minima. On the theory side, current theoretical connections between flatness and generalization could still be used for lottery tickets via a surrogate analysis (see, e.g., [Negrea et al 19 “In Defense of Uniform Convergence”](https://arxiv.org/abs/1912.04265)). More precisely, if one can get a generalization guarantee for a dense network using a flatness argument, then the sparse network generalization can be bounded by the same guarantee plus the difference in risks between the dense and the sparse network.

---

### Official Review · Reviewer_HZpf · 2022-10-23

**Confidence:** 3
**Correctness:** 4
**Technical Novelty And Significance:** 3
**Empirical Novelty And Significance:** 3
**Recommendation:** 6

**Clarity, Quality, Novelty And Reproducibility:**

* It is well written and the claim is clear
* It is novel approach and novel results


**Strength And Weaknesses:**

* Strengths
    * It provides extensive empirical results
    * The interpretation provided in this paper makes sense.

* Weaknesses
    * It is mostly empirical, and is not providing theoretical insight.
    * It compares with random pruning, but I'm curious the result for other pruning algorithms applied in an iterative manner.
    * Although the result is interesting, we still don't know why simple "magnitude-based" pruning can find such flatter directions in the error landscape.



**Summary Of The Paper:**

This paper provides the interpretation of IMP method in terms of the loss geometry, to better understand why and how IMP finds winning tickets. It provides various interesting empirical results. The main finding is that whether we find a matching network or not is related with whether the subnetworks at successive rounds are linearly connected in the loss landscape.


**Summary Of The Review:**

This is an interesting empirical work. Although we do not have sufficient theoretical explanation, it is good to share with the research community, given a proper comparison with existing iterative pruning methods.

---

> ### Author Response · Authors · 2022-11-15
> **Response to Reviewer HZpf**
>
> We thank the reviewer for their comments on our paper.  In response, we have added an additional comparison to Iterative Gradient Pruning (Fig. 11 and Fig. 12) and a summary of primary related work at the end of the introduction.  We address the reviewer’s concerns below:
>
> > It is mostly empirical, and is not providing theoretical insight.
>
> While this work is primarily an empirical investigation, it does draw connections to theory and identifies key open questions. In section 3.3, we apply a simplified theoretical model for random subspaces intersecting a target basin to pruning large-scale networks for image classification (ImageNet/ResNet50), arriving at a mechanistic explanation of how sharpness of the loss landscape limits the maximum allowed pruning ratio. The discrepancy between empirical and theoretical results also lead us to a (to the best of our knowledge) new finding—small weights are correlated with flatter directions—which opens up new theoretical questions. Empirical scientific investigations of deep learning phenomena are important as they can provide a new framework for thinking about these phenomena and identify avenues for further theoretical and algorithmic research; our work takes firm steps in this direction for IMP.
>
> > It compares with random pruning, but I'm curious the result for other pruning algorithms applied in an iterative manner.
>
> We have added additional experiments showing that the same error landscape results hold for [Iterative Gradient Pruning](https://arxiv.org/abs/2003.03033), which is another common iterative pruning algorithm (see Appendix H). We also compared the sharpness and the fractional distance of Gradient Pruning with Magnitude Pruning and Random Pruning (see Appendix H).
>
> > Although the result is interesting, we still don't know why simple "magnitude-based" pruning can find such flatter directions in the error landscape.
>
> This is indeed an extremely interesting question, but which, from initial investigation, we don’t think has a simple answer—exploring this could be an entire paper! However, since the focus of our paper is to try to provide a comprehensive understanding of how IMP finds matching sparse networks, we believe that this is out of scope for our paper (which as other reviewers have pointed out is already very full). We hope to pursue this question in future work and think that the finding itself is very valuable for the community and can inspire both algorithmic and theoretical developments.
>
> > …given a proper comparison with existing iterative pruning methods.
>
> In addition to the extended related work in the appendix, we have now added a summary of the primary related work at the end of the introduction.
>
> We would appreciate if the reviewer would let us know if the changes we have made have addressed all their concerns adequately.

---

> > ### Author Response · Authors · 2022-11-17
> > **Any Remaining Questions?**
> >
> > We are eager to address any remaining concerns you have that would make you comfortable recommending acceptance of our paper. Please let us know if there is anything that we've missed in our rebuttal since we have little time left to make revisions.

---

> > > ### Comment · Reviewer_HZpf · 2022-11-21
> > > **Reply to the author response**
> > >
> > > Thanks for the author response and running new experiments on IGP and comparing with IMP and random pruning. My concerns are addressed properly, so I maintain my score.

---

### Public Comment · ~Yihua_Zhang1 · 2023-02-23
**A Minor Mis-reference of the Figure**

Dear authors,

thanks for the great work and congratulations to the acceptance.

I was reading your paper and found that on Page 8 you said ("Fig. 5 shows the results of comparing the prediction of Lemma 3.1 to both random and magnitude pruning at a variety of pruning ratios f and levels L."). If I understand it correctly, the figure you are referring to should be Fig. 6, right? I did not see any reviewer raising this issue, so I am writing this comment for your record.

Best,

---

### Decision · Program_Chairs · 2023-01-20

**Decision:**

Accept: notable-top-25%

**Justification For Why Not Higher Score:**

Reviewers identified a few weaknesses, and the bar for oral is extremely high.

**Justification For Why Not Lower Score:**

Most of the weaknesses identified by reviewers were fixed in the revision phase; overall an impressive effort by the authors!

**Metareview: Summary, Strengths And Weaknesses:**

This paper studies lotter tickets, and connects them to many other aspects of the training process, such as SGD and flat minima.  Reviewers have a few misgivings, but may were corrected during the discussion phase, and I am glad to mark this as a clear accept.

**Note From Pc:**

if the above contains the word "oral" or "spotlight" please see: "oral" presentation means -> notable-top-5% and "spotlight" means -> notable-top-25%. As stated in our emails, we are disassociating presentation type from AC recommendations